

# A daily, 250 m, and real-time gross primary productivity product (2000 – present) covering the Contiguous United States

Chongya Jiang[1,2*], Kaiyu Guan[1,2,3*], Genghong Wu[1,2], Bin Peng[1,3], and Sheng Wang[1,2]

[1]College of Agricultural, Consumer and Environmental Sciences, University of Illinois at Urbana Champaign, Urbana, IL
61801, USA
[2]Center for Advanced Bioenergy and Bioproducts Innovation, University of Illinois at Urbana Champaign, Urbana, IL
61801, USA
[3]National Center of Supercomputing Applications, University of Illinois at Urbana Champaign, Urbana, IL 61801, USA

*Correspondence to*: Chongya Jiang (chongya.jiang@email.com) Kaiyu Guan (kaiyuguan@gmail.com)

**Abstract.** Gross primary productivity (GPP) quantifies the amount of carbon dioxide ($CO_2$) fixed by plants through photosynthesis. Although as a key quantity of terrestrial ecosystems, there is a lack of high-spatial-and-temporal-resolution, real-time, and observation-based GPP products. To address this critical gap, here we leverage a state-of-the-art vegetation index, near-infrared reflectance of vegetation ($NIR_V$), along with accurate photosynthetically active radiation (PAR), to produce a SatelLite Only Photosynthesis Estimation (SLOPE) GPP product in the Contiguous United States (CONUS).
Compared to existing GPP products, the proposed SLOPE product is advanced in its spatial resolution (250 m versus > 500 m), temporal resolution (daily versus 8-day), instantaneity (1 day latency versus > 2 weeks latency), and quantitative uncertainty (on a per-pixel and daily basis versus no uncertainty information available). These characteristics are achieved because of several technical innovations employed in this study: (1) SLOPE couples machine learning models with MODIS atmosphere and land products to accurately estimate PAR. (2) SLOPE couples highly efficient and pragmatic gap-filling and
filtering algorithms with surface reflectance acquired by both Terra and Aqua MODIS satellites to derive a soil-adjusted $NIR_V$ ($SANIR_V$) dataset. (3) SLOPE couples a temporal pattern recognition approach with a long-term Crop Data Layer (CDL) product to predict dynamic C4 crop fraction. Through developing a parsimonious model with only two slope parameters, the proposed SLOPE product explains 84% of the spatial and temporal variations in GPP acquired from 50 AmeriFlux eddy covariance sites (332 site-years), with a root-mean-square error (RMSE) of 1.65 gC $m^{-2}$ $d^{-1}$. With such a
satisfactory performance and its distinct characteristics in spatiotemporal resolution and instantaneity, the proposed SLOPE GPP product is promising for regional carbon cycle research and a broad range of real-time applications. The archived dataset is available at https://doi.org/10.3334/ORNLDAAC/1786 (Download page: https://daac.ornl.gov/daacdata/cms/SLOPE_GPP_CONUS/data/) (Jiang and Guan, 2020), and the real-time dataset is available upon request.



## 1 Introduction

Gross primary productivity (GPP) quantifies the amount of carbon dioxide ($CO_2$) fixed by plants through photosynthesis (Beer et al., 2010; Jung et al., 2017). Because GPP is the largest carbon flux and influences other ecosystem processes such as respiration and transpiration, monitoring GPP is crucial for understanding the global carbon budget and terrestrial ecosystem dynamics (Bonan, 2019; Friedlingstein et al., 2019). In addition, biomass accumulation driven by GPP is the basis for food, feed, wood and fiber production, and therefore monitoring GPP is crucial for human welfare and development (Guan et al., 2016; Ryu et al., 2019).

Over the past two decades, a number of GPP products with different spatial and temporal characteristics have been derived using remote sensing approaches (Xiao et al., 2019). However, since GPP cannot be directly observed at large scales, different models have been developed and used in generating GPP products. Process-based models use a series of nonlinear equations to represent the atmosphere-vegetation-soil system and associated fluxes. For example, a publicly-available global GPP product using process-based models is the Breathing Earth System Simulator (BESS) (Jiang and Ryu, 2016). Machine-learning models upscale site-observed GPP to a larger scale by establishing non-parametric relationships between ground-truth and gridded explanatory variables. The FLUXCOM GPP product is a typical example of this approach (Jung et al., 2019). Semi-empirical approaches utilize equations with a concise physiological meaning (e.g., light use efficiency) that are parameterized with several empirical constraint functions. The MOD17 GPP product (Running et al., 2004), the Vegetation Photosynthesis Model (VPM) GPP product (Zhang et al., 2017), and the Global LAnd Surface Satellite (GLASS) GPP product (Yuan et al., 2010) belong to this category.

With differing principles, assumptions and complexity, existing remote sensing GPP models heavily rely upon inputs with large uncertainties. First, climate forcing, such as temperature, humidity, precipitation and wind speed, are commonly used in these GPP models. However, these meteorological data are not observed but derived from general circulation models (GCMs) and usually have coarse spatial resolution (e.g., 50-km) and large time lag (e.g., weeks). Second, plant functional types (PFTs) are used to define different parameterization schemes in those models. To date, satellite land cover products are usually characterized by considerably large time lag (> 1 year), relatively low accuracy (≤ 70%) (Yang et al., 2017), and more uncertainties with regards to year-to-year variations (Cai et al., 2014; Li et al., 2018). Third, high-level remote sensing land products such as leaf area index (LAI), fraction of absorbed photosynthetically active radiation (FPAR), clumping index (CI), land surface temperature (LST) and soil moisture (SM) are used by some models. These variables are not directly observed but retrieved by complicated algorithms, and their accuracy still needs significant improvement to meet requirements of earth system models (GCOS, 2011).



Alternative approaches which heavily rely on reliable satellite observations with low dependence on uncertain model structure/parameterization and model inputs are highly required. Solar-induced fluorescence (SIF) emerged in recent years may provide a new opportunity for GPP estimation (Guanter et al., 2014). Linear relationships have been found between SIF
and GPP at various ecosystems (Liu et al., 2017; Magney et al., 2019; Yang et al., 2015). However, satellite SIF data generally have coarse resolution, large spatial gaps, short temporal coverage, and limited quality (Bacour et al., 2019; Zhang et al., 2018), and therefore not suitable for many applications.

Near-infrared reflectance of vegetation ($NIR_{V,Ref}$), defined as the product of normalized difference vegetation index (NDVI)
and observed NIR reflectance ($NIR_{Ref}$) (Eq. [1]), has recently been presented as a proxy of GPP (Badgley et al., 2017). A global monthly 0.5° GPP dataset has been produced from satellite data using the linear relationship between $NIR_{V,Ref}$ and GPP (Badgley et al., 2019), explaining 68% GPP variations observed by the FLUXNET network. Several field studies have recently found that taking incoming radiation into account further improves the $NIR_V \sim$ GPP relationship (Dechant et al., 2020; Wu et al., 2020). Because MODIS provides long-term and real-time (2000 – present) observations of red and NIR
reflectance and atmospheric conditions with high spatial (250 m for reflectance and 1 km for atmosphere) and temporal (daily) resolutions, now there is an unprecedented opportunity to generate an observation-based GPP product.

$$NIR_{V,Ref} = NDVI \times NIR_{Ref} = \frac{NIR_{Ref} - Red_{Ref}}{NIR_{Ref} + Red_{Ref}} \times NIR_{Ref} \qquad (1)$$

Leveraging the concept of $NIR_V$, here we present a new GPP model and the resultant daily, 250m, and real-time GPP product (2000 – present) covering the Contiguous United States (CONUS) (Jiang and Guan, 2020). The product is named
SatelLite Only Photosynthesis Estimation (SLOPE) because (1) the model only uses satellite data, and (2) the model only has two slope parameters. Detailed model design, multi-source satellite data processing, and comprehensive evaluation procedures are elucidated below.

## 2 Production of the SLOPE product

The method we used to estimate GPP using the novel vegetation index $NIR_{V,Ref}$ follows the concept of light use efficiency
(LUE) (Monteith, 1972; Monteith and Moss, 1977):

$$GPP = PAR \times FPAR \times LUE \qquad (2)$$

Since $NIR_{V,Ref}$ has been found strongly correlated to FPAR (Badgley et al., 2017), and moderately correlated to LUE (Dechant et al., 2019), it is possible to simplify Eq. (2) as:

$$GPP \approx PAR \times (a \times NIR_{V,Ref} + b) \qquad (3)$$

where $a$ and $b$ are slope and intercept which can be fitted from ground GPP observations. Both PAR and $NIR_{V,Ref}$ can be easily derived from satellite observations with high spatial and temporal resolutions in real time, avoiding complicated but





uncertain algorithm/parameterization to quantify FPAR and LUE in Eq. (2). This linear relationship is likely to converge within C3 species (Badgley et al., 2019), but differs between C3 and C4 species (Wu et al., 2019). Accordingly, land cover data with considerably large time lags and relatively low accuracy may not be necessary for the model parameterization. Instead, an in-season C3/C4 species dataset is needed for the accurate calibration of the linear relationship.

Defining the ratio of GPP to PAR as the incident PAR use efficiency (iPUE) gives:

$$iPUE = GPP/PAR = FPAR \times LUE \approx a \times NIR_{V,Ref} + b \tag{4}$$

Here iPUE is a confounding factor of canopy structure and leaf physiology, representing the capacity of plants to use incoming radiation for photosynthesis. When vegetation is absent, iPUE is zero and $NIR_{V,Ref}$ should be zero too. However, this is not true in reality as >99.9% soils have positive $NIR_{V,Ref}$ values according to a global soil spectral library (Jiang and Fang, 2019), and the correction of $NIR_{V,Ref}$ for soil is needed for better performance at low vegetation cover (Zeng et al.,

2019). To address this issue, we will propose spatially-explicit correction for $NIR_{V,Ref}$ to derive a soil adjusted index $SANIR_V$ (see details in section 2.2). Since $SANIR_V = 0$ when iPUE = 0, Eq. (4) becomes:

$$iPUE \approx c \times SANIR_V \tag{5}$$

where $c$ is the slope coefficient.

Considering the presence of mixed pixel of C3 and C4 species with the 250 m pixels, Eq. (5) can be rewritten as:

$$iPUE \approx [c_{C4} \times f_{C4} + c_{C3} \times (1 - f_{C4})] \times SANIR_V \tag{6}$$

where $c_{C4}$ and $c_{C3}$ are the coefficients for C4 and C3 species, respectively, and $f_{C4}$ is the fraction of C4 species in vegetation. Therefore, the SLOPE GPP model is:

$$GPP \approx [c_{C4} \times f_{C4} + c_{C3} \times (1 - f_{C4})] \times PAR \times SANIR_V \tag{7}$$

In the SLOPE model (Eq. [7]), PAR, $SANIR_V$ and $f_{C4}$ are remote sensing inputs, whereas $c_{C4}$ and $c_{C3}$ are model parameters to be calibrated using ground-truth GPP data (Fig. 1). In the following sections, we will elaborate on the derivation of PAR,

$SANIR_V$, and $f_{C4}$, along with their quantitative uncertainties, and the model calibration for parameters $c_{C4}$ and $c_{C3}$. With the uncertainty of each term ($\Delta c_{C4}$, $\Delta c_{C3}$, $\Delta f_{C4}$, $\Delta PAR$ and $\Delta SANIR_V$), the uncertainty of GPP can be estimated in a spatiotemporally-explicit manner by:

$$
\begin{aligned}
\Delta GPP \quad = \quad & (f_{C4} \times PAR \times SANIR_V) \times \Delta c_{C4} \\
+ \quad & [(1 - f_{C4}) \times PAR \times SANIR_V] \Delta c_{C3} \\
+ \quad & [(c_{C4} - c_{C3}) \times PAR \times SANIR_V] \Delta f_{C4} \\
+ \quad & \{[c_{C4} \times f_{C4} + c_{C3} \times (1 - f_{C4})] \times SANIR_V\} \Delta PAR \\
+ \quad & \{[c_{C4} \times f_{C4} + c_{C3} \times (1 - f_{C4})] \times PAR\} \Delta SANIR_V
\end{aligned}
\tag{8}
$$



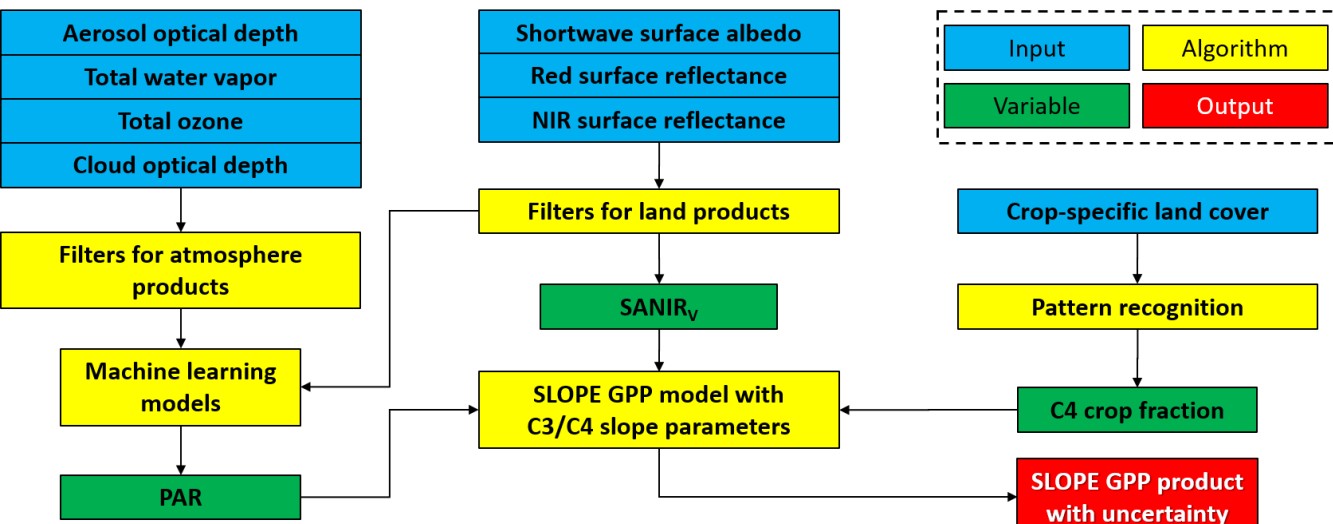

Figure 1. Framework to produce the SLOPE GPP product. The box with dash lines is the legend.

## 2.1 Derivation of PAR

SLOPE adopts several machine learning approaches to compute PAR using forcing data mainly from Terra & Aqua/MODIS Atmosphere and Land products (data solely from morning satellite Terra, afternoon satellite Aqua, and combined of the two satellites are called MOD, MYD and MCD, respectively hereinafter). The list of inputs include aerosol optical depth (AOD) at 3 km and 1 km resolutions from MOD/MYD04_3K and MCD19A2 products (Lyapustin et al., 2011; Remer et al., 2013), respectively, total column water vapour (TWV) at 1 km resolution from MOD/MYD05_L2 products (Chang et al., 2015), cloud optical thickness (COT) at 1 km resolution from MOD/MYD06_L2 products (Baum et al., 2012), total column ozone burden (TO3) at 5 km resolution from MOD/MYD07_L2 products (Borbas et al., 2015), white-sky land surface shortwave albedo (ALB) at 500 m resolution from MCD43A3 product (Román et al., 2009), and altitude (ALT) at 30 m resolution from Shuttle Radar Topography Mission Global 1 arc second (SRTMGL1) product (Kobrick and Crippen, 2017).

MODIS atmosphere products are swath data and swaths vary day by day. To maintain consistency and facilitate further usage, all data are reprojected using the nearest neighbor resampling approach to the Conus Albers projection on a NAD83 datum (EPSG: 6350) with a 1 km spatial resolution. For swath data, overlap area exists between two paths. In this case, data with smaller sensor view zenith angles provided by MOD/MYD03_L2 products are chosen. MODIS land products and SRTMGL1 are tile data with finer resolution than 1 km. They are reprojected to the EPSG 6350 spatial reference by aggregating all fine resolution pixels within each 1 km grid.

Data gaps exist in all MODIS products and gap-filling is required. For MODIS atmosphere products, gaps in MOD/MYD are first filled by data in MYD/MOD counterpart on the same day, followed by multi-year average on that day. Since the multi-





year average of COT is always non-zero, directly using it for gap-filling always implies a cloudy condition. Therefore, CLARA-2 cloud mask at 0.05° acquired from NOAA/AVHRR data is employed (Karlsson et al., 2017). Only MODIS data gaps for AVHRR cloudy pixels are filled by multi-year average of COT, whereas MODIS COT data gaps for AVHRR clear pixels are set to 0. For the MODIS land product, i.e., ALB, a temporally moving window with a 7-day radius is utilized for a specific day, and a Gaussian filter is applied to the time series data within the moving window on a per pixel basis. The

filtered values are used to fill gaps on that specific day.

Machine learning approaches are used to upscale ground-truth to satellite. Ground-truth is from the Surface Radiation Budget (SURFRAD) Network (Augustine et al., 2000), including seven long-term continuous sites across the CONUS. Daily mean shortwave radiation (SWR) and PAR on the surface are calculated from site observations at $1-3$ min intervals from

2000 through 2018. Daily mean SWR at the top of atmosphere ($SWR_{TOA}$) is calculated using latitude and day of year (DOY) information (Allen et al., 1998). Subsequently, atmospheric transmittance ($t_{SWR}$) and fraction of PAR to SW ($f_{PAR}$) are calculated as $SWR/SWR_{TOA}$ and $PAR/SWR$.

Models are built to estimate $t_{SWR}$ first, followed by $f_{PAR}$. MOD data representing atmospheric conditions in the morning and

MYD for the afternoon are used separately for the estimation, and the two estimates are averaged to account for discrepancies between morning and afternoon. Clear and cloudy conditions are also treated separately in modeling considering the absence/presence of non-zero COT data. For the estimation of $t_{SWR}$, ALB, ALT and $SWR_{TOA}$ are used in addition to atmosphere data, whereas for $f_{PAR}$, ALB, ALT and the estimated $t_{SWR}$ are used. A summary of model inputs is listed in Table 1.


Four different machine learning approaches are employed to estimate $t_{SWR}$ and $f_{PAR}$. They are least absolute shrinkage and selection operator (LASSO) (Tibshirani, 1996), multivariate adaptive regression splines (MARS) (Friedman, 1991), k-nearest neighbors regression (KNN) (Goldberger et al., 2005), and random forest regression (RF) (Liaw and Wiener, 2002). All inputs and outputs are the same for the four approaches. Four different PAR estimations are then obtained by Eq. (9), and

their ensemble mean and standard deviation are considered as the final estimation and uncertainty, respectively.

$$PAR = SWR_{TOA} \times t_{SWR} \times f_{PAR} \tag{9}$$

Table 1. Summary of machine learning model inputs for the estimation of $t_{SWR}$ and $f_{PAR}$. Daily estimations from MOD and MYD atmosphere data are averaged.

| Inputs | For daily $t_{SWR}$ estimation | | | | For daily $f_{PAR}$ estimation | | | |
| | MOD | | MYD | | MOD | | MYD | |
| | Clear | Cloudy | Clear | Cloudy | Clear | Cloudy | Clear | Cloudy |
|---|---|---|---|---|---|---|---|---|
| log(COT) | | √ | | √ | | √ | | √ |





| | | | | | | | | |
|---|---|---|---|---|---|---|---|---|
| log(AOD) | √ | √ | √ | √ | √ | √ | √ | √ |
| TWV | √ | √ | √ | √ | √ | √ | √ | √ |
| TO3 | √ | √ | √ | √ | √ | √ | √ | √ |
| ALB | √ | √ | √ | √ | √ | √ | √ | √ |
| ALT | √ | √ | √ | √ | √ | √ | √ | √ |
| $SWR_{TOA}$ | √ | √ | √ | √ | | | | |
| $t_{SWR}$ | | | | | √ | √ | √ | √ |

## 2.2 Derivation of SANIR$_v$

SLOPE derives $NIR_{V,Ref}$ (Eq. [1]) from MODIS band 1 (red) and band 2 (NIR) surface reflectance (SR) at 250 m resolution from MOD/MYD09GQ products (Vermote et al., 2002). Since cloud and cloud shadows substantially reduce $NIR_{V,Ref}$ values, SLOPE adopts three strategies to mitigate the cloud contamination.

First, the cloud mask is applied. MOD/MYD COT data processed in Section 2.1 are resampled to the same spatial reference with MOD/MYD SR data and used to mask out cloudy pixels. At this point, a morphological dilation operation is used to enlarge the cloud mask to account for cloud edges. However, since COT data have a coarser resolution (1 km) than SR data (250 m), there are still partial clouds and cloud shadows left after this step.

Second, MOD and MYD data are combined. Ideally, on a specific day, MOD and MYD $NIR_{V,Ref}$ should be identical if they are obtained under the same conditions. However, the remaining cloud contamination and sensor view zenith angle could cause differences between morning and afternoon observations. Considering vegetation index is more sensitive to cloud contamination than sensor view zenith angle, a simple criterion is applied to combine MOD and MYD observations. If the difference between MOD and MYD $NIR_{V,Ref}$ is greater than or equal to a predefined threshold (0.1 in this study), then the smaller one is likely cloud contaminated and the larger one is used. Otherwise, the average value of the two is used. However, in many cases, both MOD and MYD data are contaminated, and sensor view zenith angle may cause unexpected day-to-day variations.

Third, a temporal filter is applied. The filter is based on the assumption that $NIR_{V,Ref}$ should change smoothly within a short time period. Accordingly, a temporally moving window with a 3-day radius is utilized for a specific day. Mean and standard deviation are calculated from the $NIR_{V,Ref}$ time series on a per pixel basis. Values outside the range of mean ± 1.5 standard deviations are considered as outliers and dropped. Subsequently, the mean of the first 3 days and that of the last 3 days are calculated, respectively. If the $NIR_{V,Ref}$ value of the target day is 20% smaller or larger than both the first 3 days mean and the last 3 days mean, then that $NIR_{V,Ref}$ value is considered as an outlier and dropped.





After the removal of outliers, a large amount of data gaps exist and gap-filling is required. Similar to ALB in Section 2.1, a temporally moving window with a 3-day radius is utilized for a specific day, and a Gaussian filter is applied and used to fill gaps on that day. The rest of data gaps are filled with multi-year average of $NIR_{V,Ref}$. Considering extreme cases that no data is available on a specific day over all years, multi-year average of $\pm 3$ days is used for the final gap-filling.


To minimize the effects of variations in soil brightness on $NIR_{V,Ref}$, soil background $NIR_V$ ($NIR_{V,Soil}$) is identified from multi-year average $NIR_{V,Ref}$ time series. Three features of soil $NIR_{V,Ref}$ are utilized. First, for each pixel, soil background can be considered as an approximate constant over time. Soil moisture could remarkably influence soil brightness but marginally influence the relative difference between NIR and red reflectance. Second, soil background should have smaller $NIR_{V,Ref}$

value than vegetation, and therefore $NIR_{V,Soil}$ should be smaller than the seasonal mean $NIR_V$. Third, according to a global soil spectral library (Jiang and Fang, 2019), more than 99.99% soils have $NIR_{V,Ref}$ values smaller than 0.2. Accordingly, the $NIR_{V,Ref}$ range of [0, 0.2] is segmented into 20 bins with 0.01 for each interval. Each day in the multi-year average $NIR_{V,Ref}$ time series falls into one bin. Bins larger than the seasonal mean $NIR_V$ are excluded. The central value of the bin with the most days is set as $NIR_{V,Soil}$. Evergreen species could be exceptions that the $NIR_{V,Soil}$ value obtained from the prevailing bin

is actually $NIR_{V,Ref}$ from vegetation. To address this issue, pixels with $NIR_{V,Soil}$ value larger than 0.1 and seasonal coefficient of variation (CV) of $NIR_{V,Ref}$ smaller than 33% are empirically considered as evergreen species and their $NIR_{V,Soil}$ values are set to 0.

Finally, $SANIR_V$ is defined as:

$$SANIR_V = \frac{NIR_{V,Ref} - NIR_{V,Soil}}{NIR_{V,Peak} - NIR_{V,Soil}} \times NIR_{V,Peak} \tag{10}$$

where $NIR_{V,Peak}$ is the maximum value of multi-year average $NIR_{V,Ref}$ time series on a per-pixel basis. $SANIR_V$ does not change $NIR_{V,Peak}$, but changes more for low $NIR_{V,Ref}$ values. $SANIR_{V,Ref}$ is set 0 when $NIR_{V,Ref} \leq NIR_{V,Soil}$. In general, $SANIR_V$ is supposed to be smooth within a short time period, therefore, the standard deviation within the $\pm3$-day temporal window is calculated as uncertainty.

**2.3 Derivation of C4 fraction**

A National Land Cover Database (NLCD) along with a crop-specific land cover product Cropland Data Layer (CDL) are used to derive the fraction cover of C4 crop in vegetation ($f_{C4}$). NLCD is a comprehensive land cover database produced by the United States Geological Survey (USGS). It provides several main thematic classes such as urban, agriculture, and forest with high accuracy (Homer et al., 2004). The 30 m nationwide NLCD data are available in 2001, 2004, 2006, 2008, 2011,

2013 and 2016. CDL is an agriculture-oriented product produced by the United States Department of Agriculture (USDA). It provides > 100 crop cover types and leverages other land cover types from NLCD (Boryan et al., 2011). Across the CONUS



CDL data are available at a 30m spatial resolution and in a yearly temporal frequency from 2008 through 2019, whereas in some areas annual data are available back to the 1990s.

The fraction of C4 crop in vegetated areas is first derived using existing CDL data. NLCD land cover types are categorized into vegetated areas and non-vegetated areas with 30 m resolution. Fraction of vegetated areas in total area is subsequently calculated for each 250 m pixel. Similarly, CDL crop types are categorized into C4-planted areas and non-C4 areas with 30 m resolution. Fraction of C4 crops in total area is subsequently calculated for each 250 m pixel. The ratio of fraction of C4 crops in total area to fraction of vegetated areas in total area is calculated to derive fraction of C4 crop in vegetated areas at

250 m resolution. Since NLCD data is not available every year, an assumption is made that one year NLCD data can represent adjacent years. Specifically, NLCD 2001 is used for 2000 – 2002, NLCD 2004 is used for 2003 and 2004, NLCD 2006 is used for 2005 and 2006, NLCD 2008 is used for 2007 – 2009, NLCD 2011 is used for 2010 and 2011, NLCD 2013 is used for 2012 – 2014. NLCD 2016 is used for 2015 – 2019.

To predict the fraction of C4 crop in vegetation for region-years that no CDL data is available, crop rotation patterns are identified from historical data. Assuming that C4 crops are planted following three rotation strategies: C4/non-C4, C4/C4/non-C4, and non-C4/non-C4/C4, and assigning 1 to C4 and 0 to non-C4, a total of eight possible time series during the period of 2008 – 2019 when nationwide CDL data are available are listed in Table 2. On a per-pixel basis, the time series of the fraction of C4 crop in vegetation during 2008 – 2019 is compared with the eight predefined rotation patterns.

PeaSLOPEn coefficient $r$ is calculated between actual time series and each of the eight patterns, and the pattern yielding the largest $r$ is the identified rotation pattern. Once the pattern is identified, fraction of C4 crop in vegetated areas in any unknown year can be inferred. If one year is inferred as C4, then the multi-year average of C4 fraction over C4-dominated years is used. Otherwise, the multi-year average over C3-dominated years is used. If the largest $r$ is smaller than 0.497, i.e., $p > 0.1$ for 12 years, then it is considered as no significant pattern and the multi-year average over all years is used. The

RMSE between predicted and reference CDL C4 fraction is calculated as uncertainty. To account for the land cover change, the predicted C4 crop fraction is set to 0 in years when NLCD data is not classified as cropland. It is worth mentioning that C4 grassland and shrubland are not considered in this study as no nationwide high-resolution distribution data is available.

Table 2. Predefined C4-planting patterns from 2008 through 2019. If C4 crop dominates in a specific year, 1 is assigned.
Otherwise, 0 is assigned.

| Year | Pattern 1 | Pattern 2 | Pattern 3 | Pattern 4 | Pattern 5 | Pattern 6 | Pattern 7 | Pattern 8 |
|------|-----------|-----------|-----------|-----------|-----------|-----------|-----------|-----------|
| 2008 | 1 | 0 | 1 | 1 | 0 | 0 | 0 | 1 |
| 2009 | 0 | 1 | 1 | 0 | 1 | 0 | 1 | 0 |
| 2010 | 1 | 0 | 0 | 1 | 1 | 1 | 0 | 0 |
| 2011 | 0 | 1 | 1 | 1 | 0 | 0 | 0 | 1 |





| 2012 | 1 | 0 | 1 | 0 | 1 | 0 | 1 | 0 |
|------|---|---|---|---|---|---|---|---|
| 2013 | 0 | 1 | 0 | 1 | 1 | 1 | 0 | 0 |
| 2014 | 1 | 0 | 1 | 1 | 0 | 0 | 0 | 1 |
| 2015 | 0 | 1 | 1 | 0 | 1 | 0 | 1 | 0 |
| 2016 | 1 | 0 | 0 | 1 | 1 | 1 | 0 | 0 |
| 2017 | 0 | 1 | 1 | 1 | 0 | 0 | 0 | 1 |
| 2018 | 1 | 0 | 1 | 0 | 1 | 0 | 1 | 0 |
| 2019 | 0 | 1 | 0 | 1 | 1 | 1 | 0 | 0 |

## 2.4 Calibration for iPUE coefficients

SLOPE was calibrated using the GPP data derived from AmeriFlux site observations. The AmeriFlux network is a community of sites that use eddy-covariance technology to measure landscape-level carbon, water, and energy fluxes across

the Americas (Baldocchi et al., 2001). A total of 50 sites (332 site years) were involved in this study (Table S3). All of the 43 sites in the FLUXNET2015 Tier 1 dataset (variable name: GPP_DT_VUT_MEAN; quality control: NEE_VUT_REF_QC) in the CONUS were used, because this dataset was produced by standardized data processing pipeline with strict data quality control protocols and is commonly considered as ground-truth. Additional 7 sites were from the AmeriFlux level 4 dataset (variable name: GPP_or_MDS; quality control: NEE_or_fMDSsqc). This dataset was generated more than ten years ago and

only AmeriFlux Core Sites that are not covered by FLUXNET2015 were used for data quality consideration. For both datasets, only days with the best quality control flags were used in the SLOPE modelling and evaluation procedures.

We used Eq. (5) to conduct model calibration. Although SLOPE considers iPUE ~ SANIR$_V$ relationship for C3 and C4 species, we also tested other configurations for comparison purposes. Configuration 1 ("all"): all data were used together to

fit a universal iPUE coefficient $c$. Configuration 2 ("C3/C4"): data were separated for C3 and C4 species to fit $c_{C3}$ and $c_{C4}$, respectively. It is worth mentioning that only C4 crops (6 sites) were considered as C4 species, whereas C4 grass and shrubs (3 sites: US-SRG, US-SRM and US-Wkg) were still categorized into C3 species because of the lack of nationwide and high-resolution C4 grass/shrubs data. Configuration 3 ("PFTs"): data were separated for different PFTs: evergreen needleleaf forest (ENF; 14 sites), deciduous broadleaf forest and mixed forest (DBF & MF; 8 sites), shrubland and woody savannah

(SHR & WSA, 5 sites), grassland (GRA; 8 sites), wetland (WET; 5 sites), C3 cropland (10 sites) and C4 cropland (6 sites), to fit PFT-specific iPUE coefficients. The RMSE between SANIR$_V$-derived and AmeriFlux iPUE for C3 and C4 are calculated as uncertainties of $c_{C3}$ and $c_{C4}$, respectively.




# 3 Evaluation of the SLOPE product

## 3.1 Performance of PAR

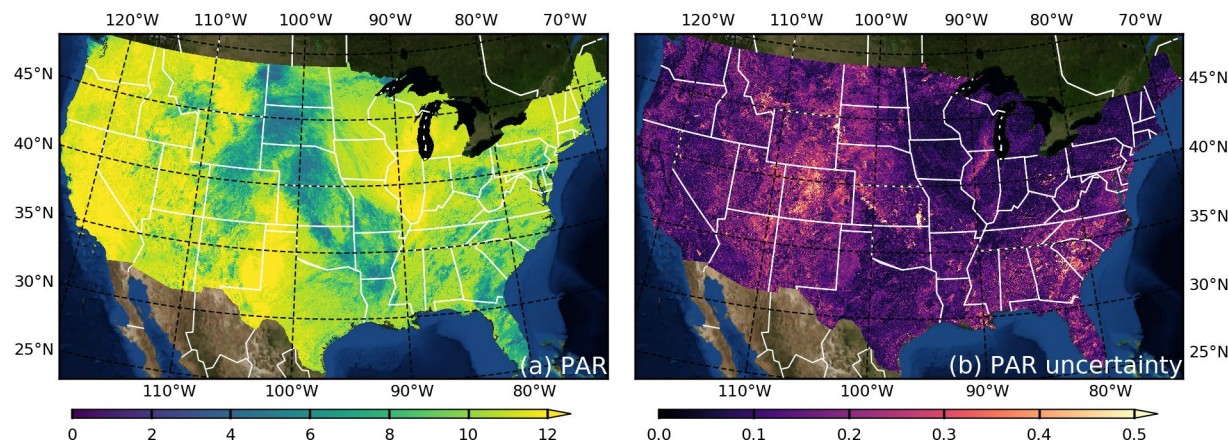


Figure 2. Spatial distribution of 1 km resolution (a) PAR (W m$^{-2}$) and (b) PAR uncertainty (W m$^{-2}$) on Aug 1, 2019. The background image is a ⓒ NASA Blue Marble image.

SLOPE PAR demonstrates distinctive and detailed spatial variations in the CONUS because of the large spatial variations of
atmospheric conditions (Fig. 2a). As an example, on Aug 1, 2019, large areas in the central and southeastern parts of the CONUS display significantly lower values than other areas, due to dominant impacts of cloud (Fig. S1) and considerable influences of water vapor in nearby cloud-free regions (Fig. S2). Aerosol optical depth also influences clear-sky PAR to some degree. For example, West Illinois and South California show little aerosol and thus higher PAR values than surrounding areas. PAR uncertainties caused by the difference of the four machine learning algorithms are generally small (<
5%; Fig. 2b). Higher uncertainties are mainly distributed in cloudy areas. The background image is a ⓒ NASA Blue Marble image.

To evaluate the SLOPE PAR, we used two different site observation datasets which are independent of the PAR derivation procedure. The first dataset is SURFRAD (Table S1). While SURFRAD data from 2000 through 2018 were used for model
training, we used data in 2019 for evaluation. The second dataset is FLUXNET2015 (Table S2). A total of 41 sites providing PAR data were used for the evaluation. For both datasets, only days with the best quality control flags were used.

Evaluation results show that SLOPE PAR is in a highly aligned agreement with ground truth independent from the training procedure (Fig. 3). Across the seven SURFRAD sites in 2019 and the 41 AmeriFlux sites from 2000 to 2014, SLOPE PAR
achieves an overall coefficient of determination ($R^2$) of 0.91, and root-mean-square errors (RMSE) of 1.09 and 1.19 MJ m$^{-2}$ d$^{-1}$, respectively.



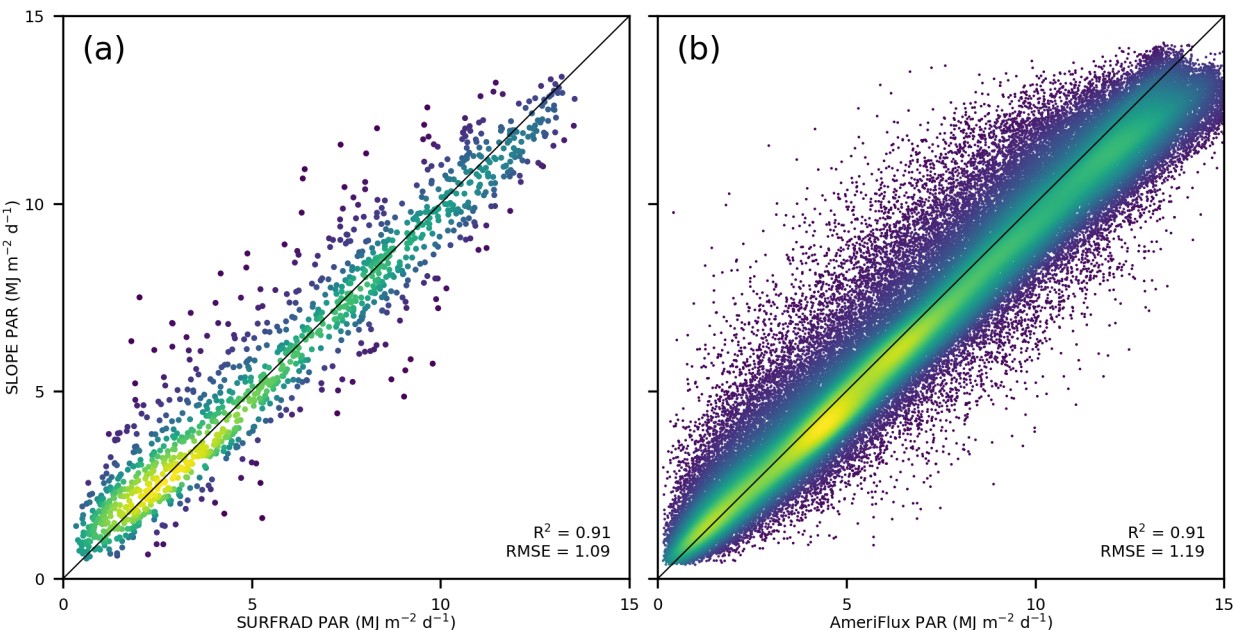

Figure 3. Comparison between site-observed PAR and SLOPE PAR. (a) Comparison across seven SURFRAD sites in 2019. (b) Comparison across 41 AmeriFlux sites from 2000 to 2014. All site data are independent of the training procedure.

## 3.2 Performance of SANIR$_V$

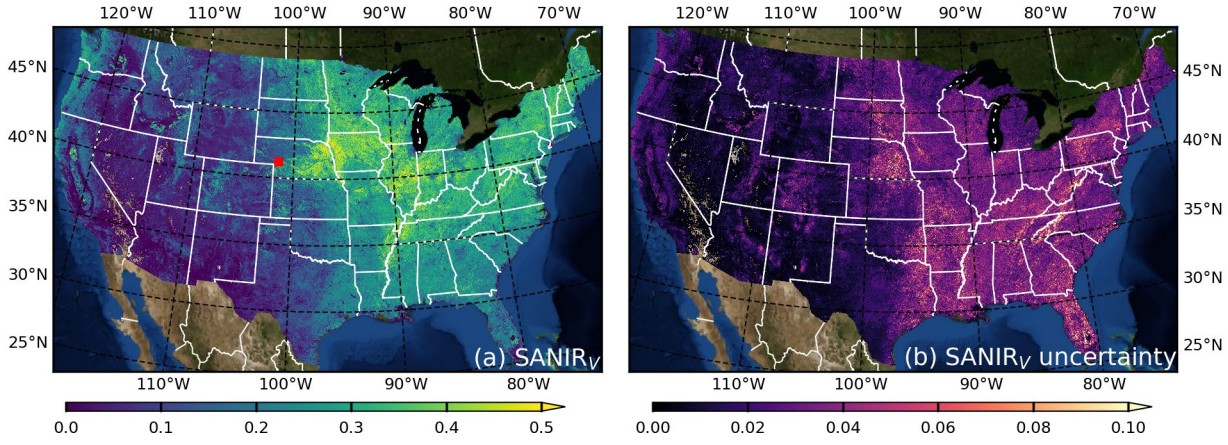



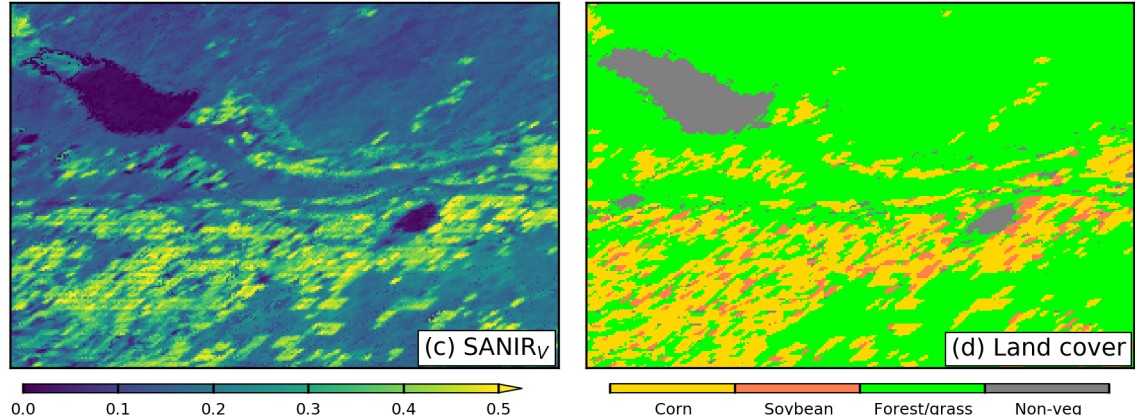

Figure 4. Spatial distribution of 250 m resolution (a and c) $SANIR_V$, (b) $SANIR_V$ uncertainty and (d) land cover on Aug 1, 2019. (c) shows a $50 \times 75$ km$^2$ area in the Keith County, Nebraska (red marker in [a]). (d) is aggregated from 30 m CDL data by selecting the dominant land cover type within each 250 m pixel. The background image is a © NASA Blue Marble image.

SLOPE $SANIR_V$ demonstrates detailed and distinctive spatial variations in the CONUS (Fig. 4a). In the peak growing season, remarkably high $SANIR_V$ values (~ 0.5) from the Corn Belt in the Central US are observed. This area is one of the most productive areas on Earth, producing more than 30% of global corn and soybean (Green et al., 2018). Forested areas in the Eastern and Western US are characterized by relatively high values (0.3 – 0.4) and medium values (0.2 – 0.3), respectively. Low values (< 0.2) are mainly observed in grasslands and shrublands in the Western US. Uncertainty is associated with $SANIR_V$ data on the pixel basis (Fig. 4b). In general, areas with higher $SANIR_V$ values also have higher uncertainties. However, this pattern is altered by atmospheric conditions, where areas with higher cloud optical thickness (Fig. S1), water vapour (Fig. S2) and aerosol optical depth (Fig. S3) values tend to have larger uncertainties. At small scale (e.g., within a county), SLOPE $SANIR_V$ also demonstrates clear spatial variations (Fig. 4c). The $SANIR_V$ values generally follow the order that soybean > corn > forest/grass > non-veg. On Aug 1, 2019, soybean was in peak growing season, whereas corn had passed, so the difference between the two crops can be observed. In addition, differences between plots can be observed, possibly indicating different varieties, planting density, and management.

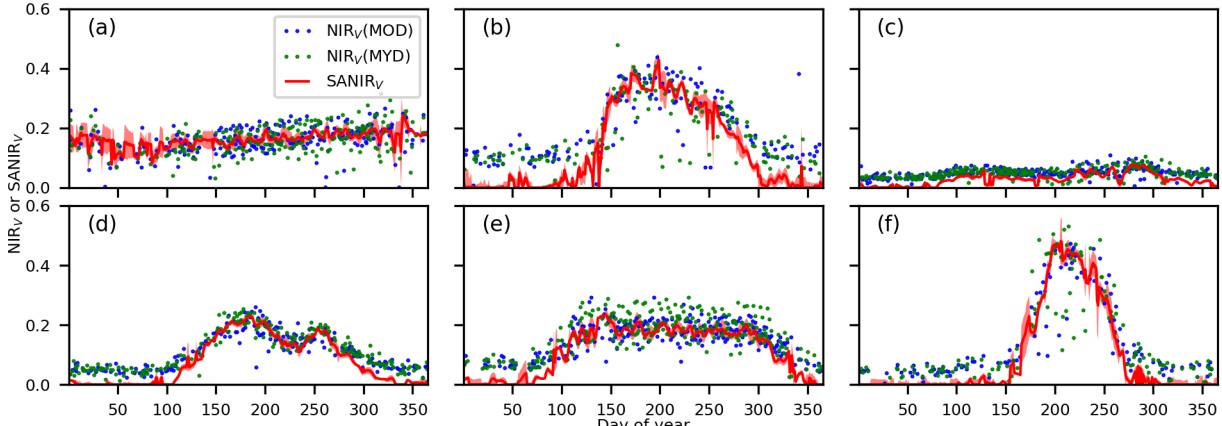

Figure 5. Comparison between SANIR$_V$ and raw NIR$_V$ derived from MOD09GQ and MYD09GQ products at six AmeriFlux sites (Table S3) in 2019. (a) US-Blo (evergreen needleleaf forest, ENF). (b) US-Ha1 (deciduous broadleaf forest, DBF). (c) US-Whs (open shrubland, OSH). (d) US-AR1 (grassland, GRA). (e) US-Myb (wetland, WET). (f) US-Bo1 (cropland, CRO). Shaded areas indicate uncertainties of SANIR$_V$.

SLOPE SANIR$_V$ shows significantly different seasonality for different PFTs (Fig. 5). The evergreen needleleaf forest site US-Blo is characterized by a relatively stable SANIR$_V$ seasonal cycle in 2019 (Fig. 5a), indicated by a CV = 14.9% only. The deciduous broadleaf forest site US-Ha1 has a large seasonal variation with a CV = 108.6% (Fig. 5b). The SANIR$_V$ value suddenly rises from 0 to 0.3 in May, reaches 0.4 in June and July, and gradually decreases back to 0 in October. The hot desert open shrubland site US-Whs has an overall low SANIR$_V$ value (Fig. 5c), with a peak value observed in early October. The grassland site US-AR1 shows a distinct double-peak (in June and September) seasonal pattern (Fig. 5d), which is caused by the precipitation seasonality there. The wetland site US-Myb is characterized by a long growing season and a flat peak from April to November (Fig. 5e). The cropland site US-Bo1 has corn planted in 2019, and it shows the highest SANIR$_V$ peak up to 0.5 among all the shown six sites (Fig. 5f). It is worth mentioning that compared to the two raw satellite-observed NIR$_V$ provided by MOD09GQ and MYD09GQ products, respectively, SLOPE SANIR$_V$ successfully removes the soil impact in the non-growing season as the values equal to or close to zero. In addition, SLOPE SANIR$_V$ is gap-free and much less contaminated by noises. Furthermore, spatiotemporally-explicit uncertainty is associated with each SANIR$_V$ value.



## 340    3.3 Performance of C4 fraction

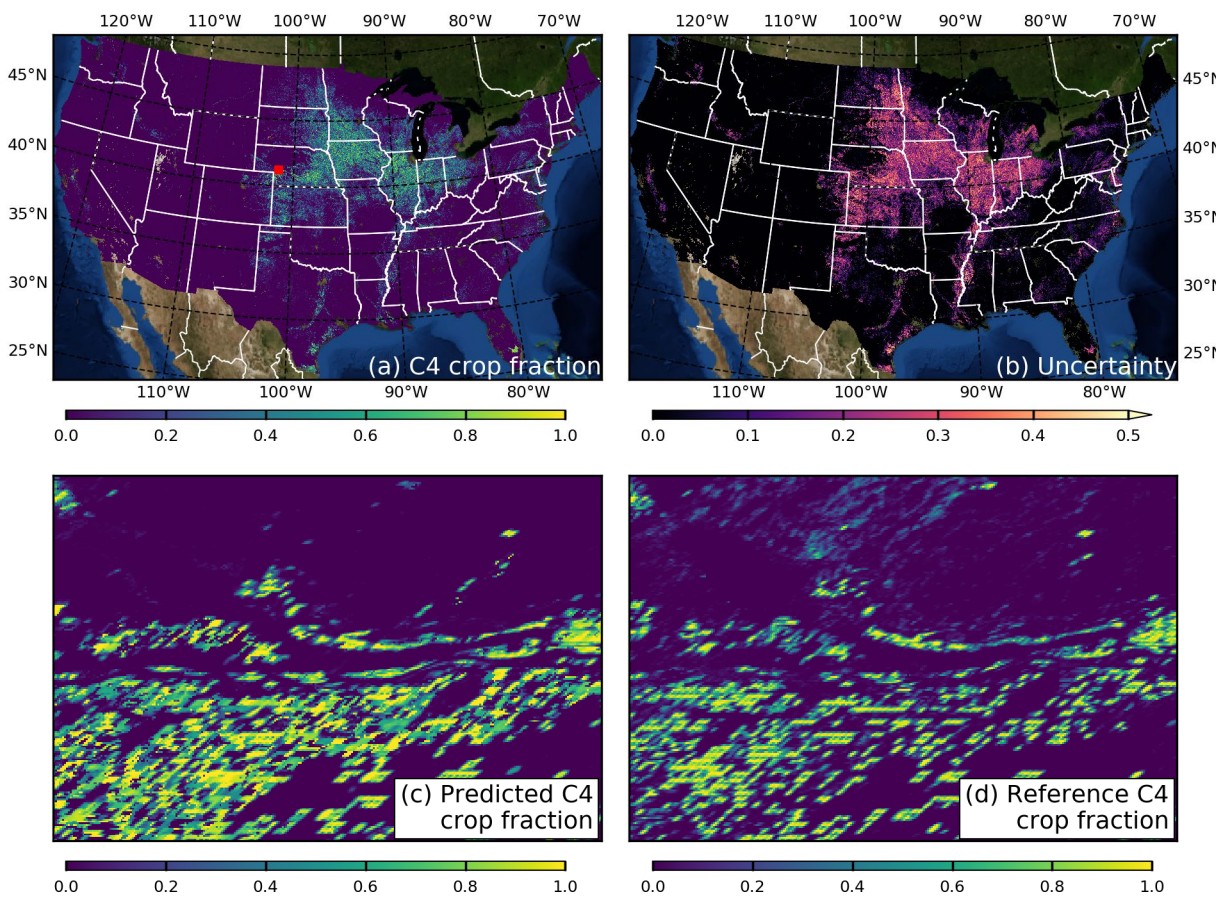

Figure 6. Spatial distribution of 250 m resolution (a and c) predicted fraction of C4 crop in vegetation in 2002, (b) C4 crop fraction uncertainty and (d) reference fraction of C4 crop in vegetation in 2000 derived from CDL. (c and d) show a 50 × 75

km² area in the Keith County, Nebraska (red marker in [a]). Only CDL data during 2008 – 2019 are used in the modelling procedure and therefore (d) is independent of (c). The background image is a ⓒ NASA Blue Marble image.

SLOPE predicts a reasonable fraction of C4 crop in vegetation in the CONUS (Fig. 6a). Most of the C4 crops are located in the Corn Belt, especially in Indiana, Illinois, Iowa and Nebraska. A direct comparison between predicted C4 crop fraction

(Fig. 6c) and independent reference CDL data (Fig. 6d) indicates that the SLOPE prediction is able to reconstruct the spatial patterns of the fraction of C4 crop in vegetation at 250 m resolution. A further investigation with regard to interannual dynamics shows that the SLOPE predictions can even perform better than CDL reference data (Fig. 7), benchmarked with ground truth collected in the field. At this point, the CDL land cover could be prone to uncertainties in both satellite observation and classification algorithm, and classification is conducted year by year without an interannual consideration

(Lark et al., 2017). SLOPE employs a rotation model to match decadal time series of CDL data, during which procedure



noises in CDL data are suppressed. The features that SLOPE is able to reconstruct spatial and interannual patterns of CDL data enables producing GPP in years when CDL data is unavailable (e.g., 2020 and years before 2008 for most regions). It is worth mentioning that uncertainty is also associated with each pixel (Fig. 6b).

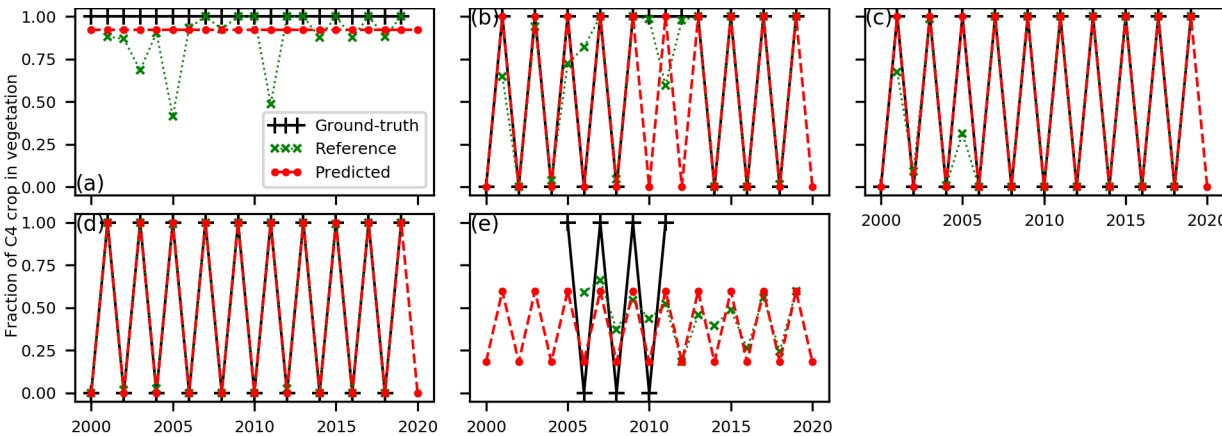

Figure 7. Comparison of fraction of C4 crop in vegetation between field collected ground truth, 250 m resolution CDL data and 250 m resolution SLOPE predictions at six AmeriFlux sites (Table S3) in the U.S. Corn Belt from 2000 to 2020. (a) US-Ne1. (b) US-Ne2. (c) US-Ne3. (d) US-Bo1. (e) US-Ro1.



## 3.4 Performance of GPP



Figure 8. Comparison between SANIR$_V$ and iPUE over different subsets. The slope value of the SANIR$_V$ ~ iPUE relationship is the model parameter $c$(Eq. [5]). (a) is used by the model configuration 1 ("all"). (b) and (c) are used by the model configuration 2 ("C3/C4") which is actually used by SLOPE. (b) and (d) – (i) are used by the model configuration 3 ("PFTs").


SLOPE SANIR$_V$ shows a strong linear correlation with iPUE (Fig. 8). When data from all 50 sites (332 site years) are used together, the SANIR$_V$ ~ iPUE relationship has an overall $R^2$ value of 0.70 (Fig. 8a). This is composed of $R^2$ of 0.91 from C4 species (Fig. 8b) and 0.66 from C3 species (Fig. 8c). C3 species can be further decomposed into six PFTs (Fig. 8d – 8i), among which cropland has the highest $R^2$ value up to 0.80 (Fig. 8i), whereas evergreen needleleaf forest has the lowest value

of 0.36 partly because of the small value ranges in both SANIR$_V$ and iPUE (Fig. 8d). The overall slope is 3.75 gC MJ$^{-1}$ for all data (Fig. 8a). Distinct difference is found between C4 (5.22; Fig.8b) and C3 (3.46; Fig.8c) species, suggesting the importance of separating C4 from C3 species in modelling. The slope values vary to a limited degree within C3 species (Fig. 8d – 8i), ranging from 3.24 gC MJ$^{-1}$ (cropland; Fig. 8i) to 3.65 gC MJ$^{-1}$ (deciduous broadleaf forest and mixed forest; Fig. 8d), indicating the insignificance of separating different PFTs. It is worth mentioning that the SANIR$_V$ ~ iPUE relationship

has a zero intercept because of the successful removal of the soil impact.

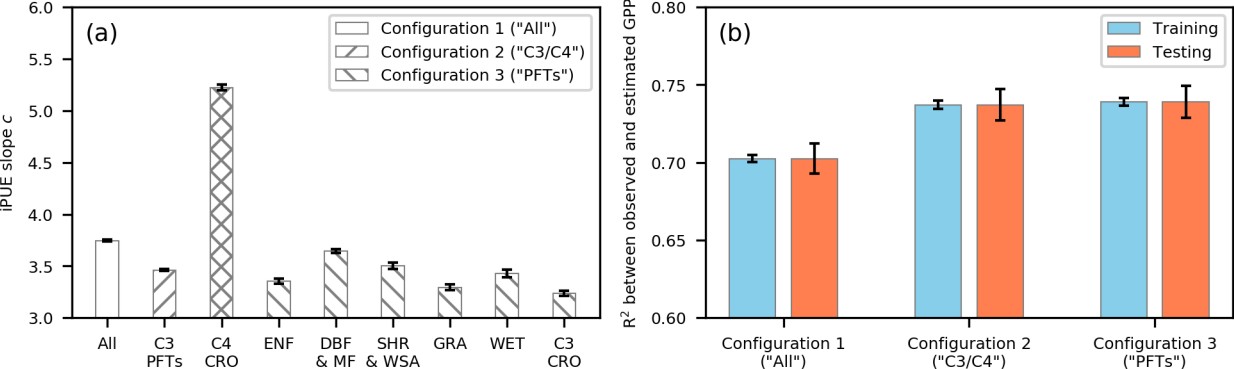

Figure 9. Statistics of the SANIR$_V$ ~ iPUE relationship from cross validation. (a) Slopes of the SANIR$_V$ ~ iPUE relationship over different subsets. (b) $R^2$ between AmeriFlux GPP and estimated GPP using different model configurations for the

training and testing datasets, respectively. Error bars in both subplots indicate 95% confidential intervals over 500 experiments.

A 100-time repeated 5-fold cross-validation reveals the robustness of the SANIR$_V$ ~ iPUE relationships (Fig. 9). Here the repeated cross-validation means the whole GPP dataset from all 50 sites (332 site years) is randomly split into 5 folds, 4

folds for training and 1 fold for testing, and the process is repeated 100 times yielding 500 training-testing splits in total. In all subsets, the uncertainties of the iPUE coefficient $c$ (the slope of the SANIR$_V$ ~ iPUE relationship) are less than 1% (Fig.



9a). When using the three different model configurations, the model performances in simulating the whole training/testing datasets also show little variation (Fig. 9b), in general < 0.5% and < 1.5% for the training and testing datasets, respectively. Moreover, the $R^2$ values between training and testing datasets, and between C3/C4 and PFT-based configurations are almost

identical ($\sim 0.74$). These results suggest using $c_{C4} = 5.22$ (Fig. 8b) and $c_{C3} = 3.46$ (Fig. 8c) in SLOPE is reasonable. The 95% confidential intervals of $c$ for C4 and C3 species (Fig. 9a) are used as their uncertainties in SLOPE.

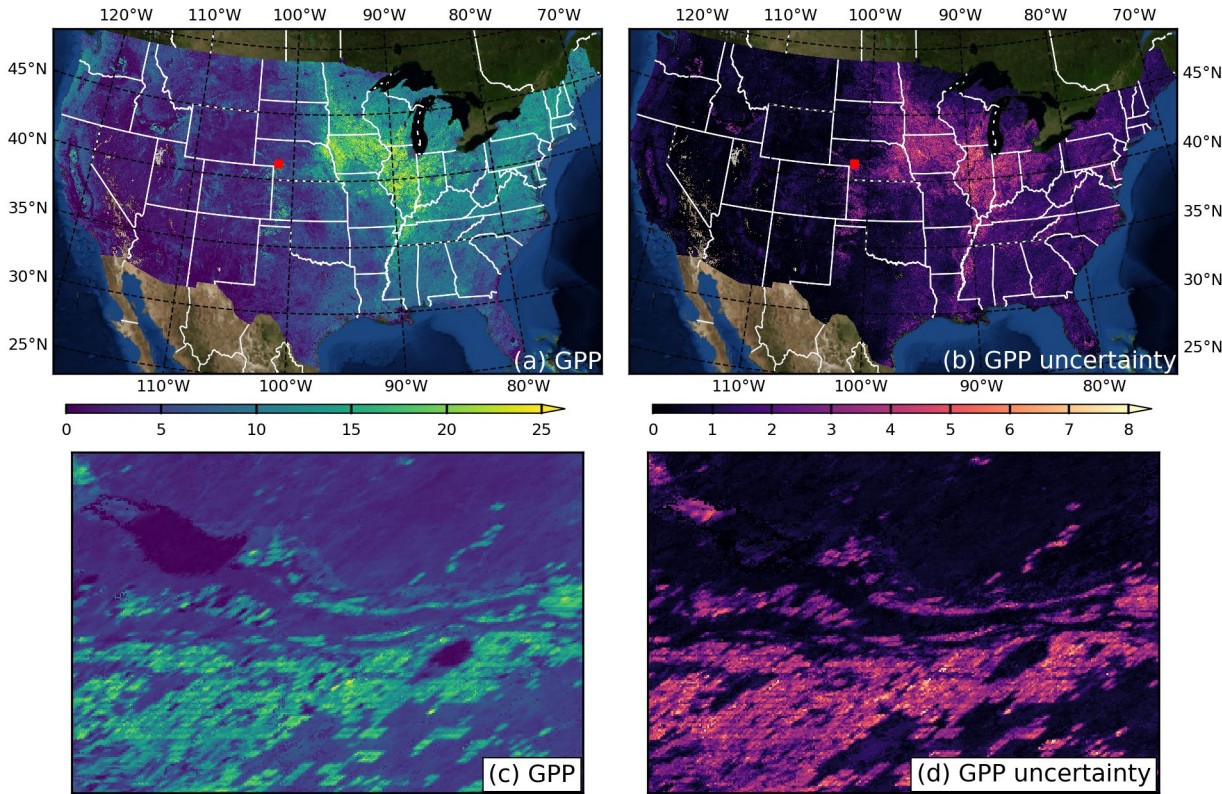

Figure 10. Spatial distribution of 250 m resolution (a and c) GPP (gC m$^{-2}$ d$^{-1}$) and (b and d) GPP uncertainty (gC m$^{-2}$ d$^{-1}$) on Aug 1, 2019. (c and d) show a 50 × 75 km$^2$ area in the Keith County, Nebraska (red marker in [a] and [b]). The background image is a © NASA Blue Marble image.

SLOPE GPP demonstrates detailed and distinctive spatial variations in the CONUS (Fig. 10a). The Corn Belt is the most

productive area, largely contributed by the C4 crop corn whose GPP could reach up to 30 gC m$^{-2}$ d$^{-1}$ (Fig. 10c and Fig. 4d). Forested areas in the Eastern US show medium GPP values, followed by forests and croplands in the Western US. Grasslands and shrublands in the Central and Western US generally show low productivity. On this example day, the $R^2$ of spatial patterns between GPP and SANIR$_V$, GPP and C4 fraction, and GPP and PAR across the CONUS are 0.89, 0.34, and

0.01, respectively. SANIR$_V$, an integrated vegetation index containing information of both FPAR and LUE (Eq. [4]),
explains the majority of GPP spatial variation. C4 fraction mainly contributes to the distribution and magnitude of the peak
in GPP spatial variation. Although PAR does not influence the nationwide GPP spatial variation, it regulates GPP values at
local scale. For example, Eastern Nebraska shows smaller GPP values than Western Iowa in spite of similar SANIR$_V$ (Fig.
4a) and C4 fraction (Fig. 6a) because of smaller PAR values (Fig. 2a). At small scale (e.g., within a county), the 250 m
resolution (~0.06 km$^2$ per pixel) SLOPE GPP is close to revealing field-level heterogeneity (Fig. 10c), considering that the
mean and median crop field sizes in the CONUS are 0.19 km$^2$ and 0.28 km$^2$, respectively (Yan and Roy, 2016). This makes a
big difference from existing global GPP products whose spatial resolutions are at least 500 m (~0.25 km$^2$ per pixel).
Quantitative uncertainty is provided for each SLOPE GPP estimate (Eq. [8]). The spatial pattern shows that the Corn Belt
has the largest uncertainty (Fig. 10b; e.g., 5 gC m$^{-2}$ d$^{-1}$) due to the considerable contribution from the uncertainty of C4
fraction (Fig. 6b).


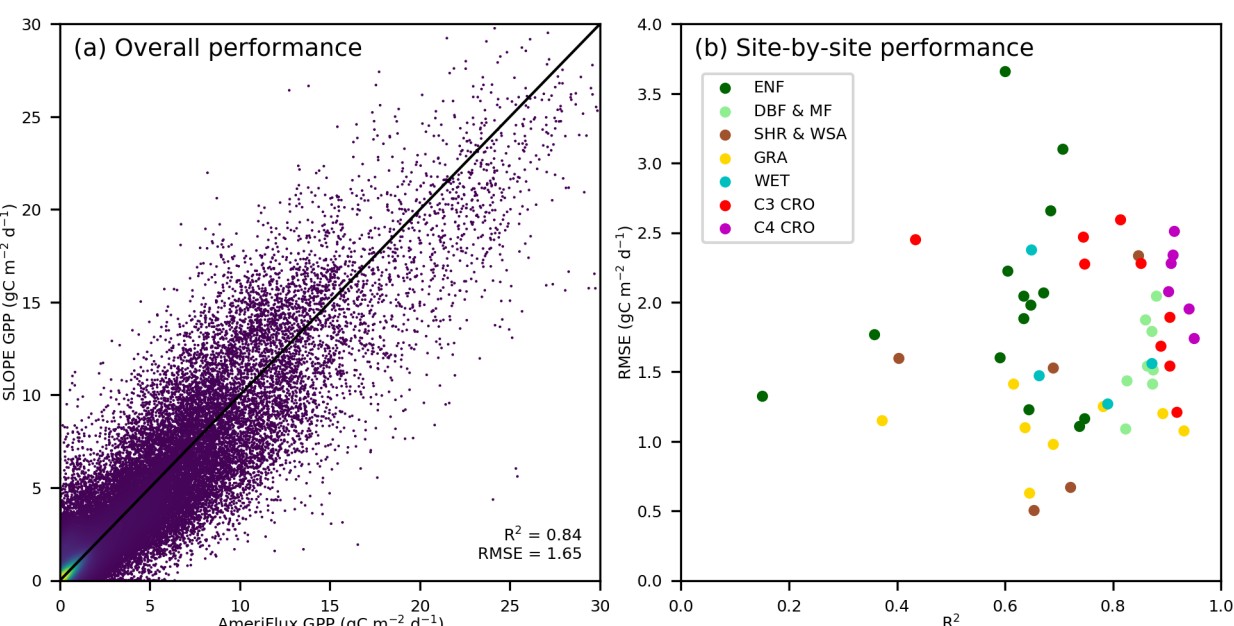

Figure 11. Performance of the SLOPE GPP. (a) Comparison between AmeriFlux GPP and SLOPE GPP across all sites. (b)
R$^2$ and RMSE of individual sites. Sites with a C3/C4 rotation are separated into C3 CRO and C4 CRO.

SLOPE GPP agrees fairly well with ground-truth from the AmeriFlux (Fig. 11). Across all of the 50 sites (332 site years; Fig.
11a), SLOPE GPP achieves an overall R$^2$ of 0.84, RMSE of 1.65 gC m$^{-2}$ d$^{-1}$, and relative error of 39.6%. For individual sites
(Fig. 11b), the median R$^2$ and RMSE are 0.75 and 1.65 gC m$^{-2}$ d$^{-1}$, respectively. C4 cropland generally shows the highest
median R$^2$ value (0.91), followed by deciduous broadleaf forest and mixed forest (0.87) and C3 cropland (0.85). The lowest
median R$^2$ value (0.64) is observed for evergreen needleleaf forest. With regard to RMSE, smaller median values are found



in grassland (1.13 gC m$^{-2}$ d$^{-1}$), wetland (1.52 gC m$^{-2}$ d$^{-1}$), shrubland and woody savannah (1.53 gC m$^{-2}$ d$^{-1}$), and deciduous broadleaf forest and mixed forest (1.53 gC m$^{-2}$ d$^{-1}$), whereas C3 (2.28 gC m$^{-2}$ d$^{-1}$) and C4 (2.18 gC m$^{-2}$ d$^{-1}$) cropland tend to have larger RMSE values.

Figure 12. Comparison between AmeriFlux (black dots) and SLOPE (red curves) daily GPP at six AmeriFlux sites (Table S3)
from 2000 to 2019. (a) US-Blo (evergreen needleleaf forest, ENF). (b) US-Ha1 (deciduous broadleaf forest, DBF). (c) US-





Whs (open shrubland, OSH). (d) US-AR1 (grassland, GRA). (e) US-Myb (wetland, WET). (f) US-Bo1 (cropland, CRO). Shaded areas indicate uncertainties of SLOPE GPP.

SLOPE GPP generally captures seasonal and interannual variations of AmeriFlux GPP for different PFTs (Fig. 12). At the
evergreen needleleaf forest site US-Blo (Fig. 12a), the GPP seasonal cycle is mainly driven by PAR as the iPUE indicated by SANIR$_V$ is fairly stable (Fig. 5a). At the deciduous broadleaf forest site US-Ha1 (Fig. 12b), the start of season and the end of season agree well between AmeriFlux GPP and SLOPE GPP. At the open shrubland site US-Whs (Fig. 12c), the quick rise and drop of GPP in response to the start and end of the wet season are clearly observed in SLOPE GPP. Even the double-peak pattern in 2011 can be observed in SLOPE GPP. At the grassland site US-AR1 (Fig. 12d), the impact of a severe
drought in the Southern Great Plains in 2011 is distinct in SLOPE GPP, as the GPP values in 2011 are only about half of those in 2010 and 2012. At the cropland site US-Bo1 (Fig. 12f), the rotation-caused year-to-year variation is distinct, indicated by higher values in odd number years with C4 crop corn planted and lower values in even years with C3 crop soybean planted (Fig. 7d). It is also observed the lowest GPP peak in 2012 when a severe drought attacked the Central US.

## 4 Data availability and data format

The archived daily 250 m resolution SLOPE GPP data product from 2000 to 2019 is distributed under a Creative Commons Attribution 4.0 License. It is publicly available at the NASA's Oak Ridge National Laboratory Distributed Active Archive Center (ORNL DAAC) with a DOI https://doi.org/10.3334/ORNLDAAC/1786 (Download page: https://daac.ornl.gov/daacdata/cms/SLOPE_GPP_CONUS/data/) (Jiang and Guan, 2020). Data from 2020 are available from the authors upon request. All data are projected in the standard MODIS Land Integerized Sinusoidal tile map projection and
are stored in GeoTIFF format files with a data type of signed 16-bit integer. Each processing tile is in size of 4800 pixels by 4800 pixels, representing approximately 1200 km by 1200 km land region. In addition to the GPP product, SLOPE PAR, SANIR$_V$, and C4 fraction, along with their uncertainties, are also released. These datasets are also stored in the same spatial projection and file format with the GPP dataset. PAR (resampled from 1 km to 250 m to be consistent with GPP) and SANIR$_V$ are provided on a daily basis, whereas C4 fraction is provided on an annual basis. A README file is provided
along with the SLOPE product, which instructs the usage of the data.

## 5 Conclusions

This study produces a long-term and real-time (2000 – present) GPP product with daily and 250 m spatial and temporal resolutions. The product is based on remote sensing only (SLOPE) model, which uses accurate PAR, soil-adjusted NIR$_v$, and dynamic C4 fraction as inputs. Evaluation against AmeriFlux ground-truth GPP shows that the SLOPE GPP product has a
reasonable accuracy, with an overall R$^2$ of 0.84 and RMSE of 1.65 gC m$^{-2}$ d$^{-1}$. To demonstrate the real-time capacity of the

SLOPE GPP product, the latest GPP data on Mar 29, 2020, one day prior to the submission of this manuscript, is shown in Fig. S4. The spatiotemporal resolution and instantaneity of the SLOPE GPP product are higher than existing global GPP products, such as MOD17, VPM, GLASS, FLUXCOM and BESS. We expect this novel GPP product can significantly contribute to various researchers and stakeholders in fields related to the regional carbon cycle, land surface processes,
ecosystem monitoring and management, and agriculture. The approaches used in this study, in particular, the derivation of SANIR$_V$, can also be applied to any other satellite platforms with the two most classical bands: red and NIR. For example, SaTallite dAta IntegRation (STAIR) Landsat-MODIS fusion data which has daily, 30 m spatiotemporal resolution and can be applied at large scale (Jiang et al., 2019; Luo et al., 2018), commercial Planet Labs data with a daily interval and spatial resolution up to 3m (Houborg and McCabe, 2016; Kimm et al., 2020), and the Advanced Very High Resolution Radiometer
(AVHRR)  with a temporal coverage as far back as 1982 (Franch et al., 2017; Jiang et al., 2017).

**Acknowledgements**

C.J., K.G., G.W. and S.W. are funded by the DOE Center for Advanced Bioenergy and Bioproducts Innovation (U.S. Department of Energy, Office of Science, Office of Biological and Environmental Research under Award Number DE-
SC0018420). Any opinions, findings, and conclusions or recommendations expressed in this publication are those of the author(s) and do not necessarily reflect the views of the U.S. Department of Energy. K.G. and B.P. are funded by NASA awards (NNX16AI56G and 80NSSC18K0170). K.G. is also funded by NSF CAREER Award (1847334). C.J. and K.G. also acknowledge the support from Blue Waters Professorship from National Center for Supercomputing Applications of UIUC. This research is part of the Blue Waters sustained-petascale computing project, which is supported by the National Science
Foundation (awards OCI-0725070 and ACI-1238993) and the state of Illinois. Blue Waters is a joint effort of the University of Illinois at Urbana-Champaign and its National Center for Supercomputing Applications. We thank NASA freely share the MODIS products.

**Author contributions**

C.J. and K.G. designed the project and the workflow. C.J. and G.W. developed the SLOPE model. C.J. processed the data and generated the GPP product. C.J., B.P. and S.W. interpreted the results and refined the experiments. C.J. wrote the manuscript, and K.G., G.W., B.P. and S.W. all contributed to the improvement of the manuscript.

**Competing interests**

The authors declare that they have no conflict of interest.



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
