# Peer review of "A daily, 250 m, and real-time gross primary productivity product (2000 – present) covering the Contiguous United States"

_Earth System Science Data, 2020_

## Referee Comment (RC1) · Anonymous Referee #1 · 3 Jul 2020

A daily, 250 m, and real-time gross primary productivity product (2000 – present) covering the Contiguous United States

Jiang et al.

The authors develop a MODIS driven light use efficiency model, that derives from new estimates of PAR and C3/C4 fraction. The authors use a soil adjusted estimate of NIRv, the near-infrared reflectance of vegetation, to determine iPUE, "incident Par use efficiency." The approach seems technically sound and the advantages of providing a high resolution estimate of GPP are quite apparent. I, however, have concerns about the reproducibility of the model and its associated uncertainties.

[Figure]

The authors have wrapped three separate innovations into a single paper. First, a new approach to estimating PAR from MODIS. Second, a new approach for estimating C3/C4 fraction. And third, combining those two new products with estimates of NIRv to derive GPP.

For the PAR modeling, either more information is needed about the "four machine learning approaches" or the authors need to provide the code used to fit these models. Each algorithm has a multitude of adjustable parameters. How were these parameters determined, what are their values, and how might future researchers modify and/or improve upon the approach? It is also unclear to me how "uncertainty" is calculated. As written (L160), it appears uncertainty is represented in terms of model-model disagreement, as opposed to model-data disagreement. This seems inappropriate, though I know that the approach has been used in other parts of this literature. Trying to wrap my head around the uncertainty terms was complicated by the fact that Figure 2 reports PAR in W m2, while Figure 3 uses MJ m2 d1. Using common units throughout would be helpful.

Some other minor issues on the PAR estimation. Equation 9 uses the term "fPAR", which in this literature often means 'fraction of photosynthetically active radiation absorbed by plants.' (e.g, their equation 4). It's quite confusing to distinguish between fpar and FPAR. Finally, the authors might consider a supplemental figure showing patterns of PAR/SW — this is a fairly well-studied, physically grounded ratio. It might also help to show that the approach works just fine in semi-cloudy (e.g, when there is more diffuse PAR) conditions.

For C3/C4, how are these uncertainties propagated into the final reported GPP estimates? From Figure 6b, it seems uncertainty is often quite high (e.g., > 40 percent). Accounting for this uncertainty seems important. For Figure 6 and 7, it could be helpful to change "Reference" to "CDL Reference" or something of that sort. At first, I was confused by the difference between "Reference" and "Ground Truth."
Perhaps most importantly, the authors mostly ignore the poor performance of their approach at evergreen needleleaf sites. Figure 8D indicates that SANIRv is not a good predictor of daily iPUE at ENF sites. The result is mentioned in L374, but relatively little discussion is offered for why this might be the case or what should be done about it. From Figure 11b, it seems that combining uncertain estimates of iPUE with PAR somewhat alleviates the poor performance within ENF, though the ENF site with an R2 of less than 0.2 stands out.

A related, global scale effort to relate NIRv to GPP [Badgley, Anderegg, Berry, Field, GCB2019] that the authors cite, identified 'deciduous' vs. 'evergreen' as being a critical parameter for model performance. Recognizing the difference in scales of the two analysis and the authors' stronger focus on C3/C4, it still feels necessary for a richer discussion of the performance of the model at ENF sites, especially given that the ultimate goal of the manuscript is to distribute a GPP dataset that researchers from across disciplines might find useful. From the analyses, it seems individuals working in agricultural contexts might find the data more reliable than those working in ENF systems. These caveats should be clearly flagged for the reader and the authors might benefit the research community by offering some discussion about what they think is going on and what future efforts might address such uncertainties.

Finally, I find it perplexing that the manuscript lists five authors but the underlying data product only lists two authors. The author contributions indicate that author G.W. helped develop the SLOPE model and that authors B.P. and S.W. were involved in refining/interpreting how the model works. Doesn't the resulting dataset and the citations it might one day receive rely on the contributions of these three authors as well?

---

## Referee Comment (RC2) · Anonymous Referee #2 · 6 Jul 2020

This manuscript details a new SateLite Only Photosynthesis Estimation (SLOPE) gross primary productivity (GPP) product based on: 1) near-infrared reflectance of vegetation (NIRv); 2) photosynthetically active radiation (PAR); and 3) C3/C4 fractional cover. The new product explains 84% of the spatial and temporal variance in GPP obtained from 50 Ameriflux eddy covariance flux tower (EC) sites. Critically, the product includes uncertainty estimates at the pixel-level, an important advance over most existing products.

Overall, I feel the paper is well written and the data product is of significant value, but perhaps mostly as an improved proxy for cropland productivity. The authors make

importance advances over previous efforts by incorporating satellite-based NIRv and PAR and removing dependence on reanalysis-based weather data. The use of C3 and C4 fractional cover is appropriately done for croplands, but importantly it does not appear appropriately handled for natural ecosystems. The work adapts the commonly used light use efficiency framework, but removes all biophysical constraint logic (e.g., response to temperature, water, nutrient limitation), and instead makes the assumption that NIRv adequately captures these constraints. While this might be a fair assumption for herbaceous and deciduous dominated ecosystems, it is likely problematic for natural evergreen dominated ecosystems. Additionally, independent of vegetation type, it is unclear if NIRv is capable of capturing changes in LUE such that $CO_2$ fertilization effects are accurately represented in this product. Further, the authors utilize a classification that separates C3 and C4 vegetation functional types, however, their input data does not separate natural C3 and C4 grasslands, which is likely problematic for western US ecosystems (which are under-represented by eddy covariance flux towers, and thus the product is not well evaluated across these regions). In my view, these critical issues need to be fully addressed before this manuscript can be considered for publication.

Major Comments: 1. Line 50-60: Some valid points are made here. However, for the CONUS region in particular, there have been previous advances that already address many of these limitations. In particular, Robinson et al. (2018) utilized high quality weather data interpolated from dense weather station networks across the region and improved landcover data from the National Landcover Data Layer (NLCD). This is a much more appropriate data product to compare the new product against and, if it's available, I recommend this comparison.

Robinson, N.P., Allred, B.W., Smith, W.K., Jones, M.O., Moreno, A., Erickson, T.A., Naugle, D.E., Running, S.W. Landsat 30 m and MODIS 250 m derived terrestrial primary production for the conterminous United States. 2018. Remote Sensing in Ecology and Conservation DOI: 10.1002/rse2.74.

[Figure]

2. Sections 3.1 and 3.2: I commend the authors for their work to provide robust uncertainty estimates for PAR and SANIRv, and subsequently SLOPE GPP estimates. This is an important advance over most previous efforts. Is the uncertainty due to PAR, SANIRv, functional classification, and statistical fit included as separate data layers with the final product? This would be a useful addition for understanding pixel-level uncertainty. Also it would be useful to see a map that shows the dominant source of uncertainty at the pixel level, which would highlight potentially how uncertainty varies by region.

3. Section 3.2: There is very low variance in SANIRv for evergreen vegetation (Figure 5), which results in the lowest correlation with EC GPP data. There are also a few ENF sites with very low correlation (Figure 11). This has been a well-known issue of vegetation reflectance-based indices, such as NIRv. Below are a number of papers that discuss this issue and all indicate SIF could be a major improvement. I recommend at minimum that this issue and ways forward, such as downscaling TROPOMI SIF (Turner et al., 2019), be included prominently in the discussion of this paper.

TS Magney, DR Bowling, BA Logan, K Grossmann, J Stutz, PD Blanken, et al. 2019 Mechanistic evidence for tracking the seasonality of photosynthesis with solar-induced fluorescence. Proceedings of the National Academy of Sciences, 116, 11640-11645

Smith, W.K, Biederman, J.A., Scott, R.L., Moore, D.J.P., He, M., Kimball, J.S., Yan, D., Hudson, A., Barnes, M.L., MacBean, N., Fox, A., Litvak, M.E. Chlorophyll Fluorescence Better Captures Seasonal and Interannual Gross Primary Productivity Dynamics Across Dryland Ecosystems of Southwestern North America. 2018. Geophysical Research Letters DOI: 10.1002/2017GL075922.

AJ Turner, P Köhler, TS Magney, C Frankenberg, I Fung, RC Cohen. 2019. A double peak in the seasonality of California's photosynthesis as observed from space. Biogeosciences 17, 405-422

4. Section 3.3: There are dynamic mixtures of C3 and C4 species throughout the

natural ecosystems of the Western US. As far as I can tell this analysis and it's reliance on NLCD data is unable to account for these important ecosystems since NLCD does not represent C3 and C4 grasslands. These are also regions where EC sites are not well represented and thus the product 1) does not accurately capture; and 2) is not well constrained or evaluated across these ecosystems. At minimum, this needs to be pointed out very clearly throughout the methods and discussion of this paper. Alternatively, and maybe more appropriately, it seems the authors present an advanced cropland productivity product and perhaps natural regions including C3/C4 grasslands and evergreen forests should be masked out. It is my view that the current work has major limitations for accurately representing natural ecosystems.

Yan, D., de Beurs, K.M. 2016. Mapping the distributions of C3and C4 grasses in the mixed-grassprairies of southwest Oklahoma using the Random Forestclassification algorithm. International Journal of Applied Earth Observation and Geoinformation 47, 125-138.

5. The authors cite Badgley et al. (2019) as justification for separating the model based on only C3 and C4 functional types. Yet, that paper also indicated that the best model fit between NIRv and GPP included separation between deciduous and evergreen ecosystem types as well. Based on this, the authors should include further justification for their model framework.

Badgley, G., Anderegg, L.D., Berry, J.A. and Field, C.B. 2019. Terrestrial Gross Primary Production: Using NIRv to Scale from Site to Globe. Global Change Biology DOI:10.1111/gcb.14729.

6. I recommend a map of the flux sites utilized overlaying the NLCD / CDL data utilized. This would highlight that the product has not been evaluated across important western US ecosystem types including dry herbaceous, shrub, and evergreen forests. Also for Table S1, one of the only dryland evergreen forest sites utilize (NR1) is reproduced twice and it's unclear if years 2000-2007 or 2000-2014 are utilized in the analysis.

[Figure]

The below referenced paper covers the major challenges associated with extrapolating algorithms across drylands without appropriate evaluation.

Smith, W.K., Dannenberg, M.P., Yan, D., Herrmann, S., Barnes, M.L., Barron-Gafford, G.A., Biederman, J.A., Ferrenberg, S., Fox, A.M., Hudson, Knowles, J.F., MacBean, N., Moore, D.J.P., Nagler, P.L., Reed, S.C., Rutherford, W.A., Scott, R.L., Wang, X., Yang, J. 2019. Remote sensing of dryland ecosystem structure and function: Progress, challenges, and opportunities. Remote Sensing of Environment 233, 111401.

Minor Comments: 1. Consider dropping the NASA Blue Marble background from all maps. This is unnecessary and potentially distracting. 2. Why use a soil adjusted NIRv? One of the advantages of NIRv is that is naturally isolates a pure vegetation signal (Bagdley et al., 2019). 3. I recommend including in the discussion whether this model is capable of capturing $CO_2$ fertilization effects on vegetation productivity. Previous work has suggested that LUE models may not have the capacity to fully capture this important and rapidly changing driver of GPP. Pointing out this potential limitation is important to ensure appropriate data usage by the community.

---

## Referee Comment (RC3) · Anonymous Referee #3 · 1 Aug 2020

This study proposes a new diagnostic estimation of gross primary productivity at a continental scale using remote sensing data. The proposed method is good for many researchers, etc., since it provides high spatial (250m) and temporal (daily) resolutions. Another advantage of this approach is its simplicity (a very small number of parameters). Also, model description, evaluation, and data availability were well-written, I believe. I have some minor comments (and questions).

Uncertainties estimation. Adding uncertainty information is very nice, and one of the most important contributions of this study. I found clear descrption on GPP uncertainties, however, it is not easy to find information on each input parameters. Please add

information how to define uncertainties in each input data.

L52-53. As far as I heard, there are several new reanalysis datasets for meteorological datasets with higher spatial resolutions. Please update the description here. I think this approach is not based on GCM, but reanalysis (of course, reanalysis is based on climate model, but reanalysis is more appropriate word. Please modify.

L77. Maybe no explanation of a symbol, REDref. Please check the manuscript.

L97. 'When vegetation in absent, iPUE is zero, and NIRvref should be zero." I don't think NIRref should be zero, since soil shows different NIRvref value. Suggest rewording.

L166-169. For quality control of surface reflectance data, did you use information on MOD09(MYD09) quality flag information? I could not find the description. If no, why?

L176. Sensor view angle is a cause of Terra, Aqua reflectance differences. Solar zenith and azimuth angles are also important as well as sensor view angle?

L196-207. Not clear, please improve it to understand it more easily.

---

## Author Comment (AC1) · 1 Oct 2020

Thank you for reviewing our manuscript "A daily, 250  m, and real-time gross primary productivity product (2000–present) covering the Contiguous United States". We have tried our best to address your comments and to improve our manuscript.

1. The authors have wrapped three separate innovations into a single paper. First, a new approach to estimating PAR from MODIS. Second, a new approach for estimating C3/C4 fraction. And third, combining those two new products with estimates of NIRv to derive GPP.

[Figure]

We greatly appreciate your positive summary on our innovations.

2. For the PAR modeling, either more information is needed about the "four machine learning approaches" or the authors need to provide the code used to fit these models. Each algorithm has a multitude of adjustable parameters. How were these parameters determined, what are their values, and how might future researchers modify and/or improve upon the approach? It is also unclear to me how "uncertainty" is calculated. As written (L160), it appears uncertainty is represented in terms of model-model disagreement, as opposed to model-data disagreement. This seems inappropriate, though I know that the approach has been used in other parts of this literature. Trying to wrap my head around the uncertainty terms was complicated by the fact that Figure 2 reports PAR in W m2, while Figure 3 uses MJ m2 d1. Using common units throughout would be helpful. We totally agreed with your points.

For the modeling, we have added the following text to the manuscript: "We used Scikit-learn, a free software machine learning library for the Python programming language, to build the models. All the four algorithms were automatically optimized by tuning their hyperparameters using five-fold-cross-validation on their training dataset." For the uncertainty, we suppose product uncertainty information can be categorized into two types: theoretical and physical (Fang et al., 2012). While physical uncertainties indicate the departure of product values from hypothetical true values and are obtained through independent validation studies, i.e., model-data disagreement, theoretical uncertainties are caused by uncertainties in the input data and model imperfections, e.g., model-model disagreement, and are usually estimated by individual product science teams. The unit in Figure 2 is a typo, and we have corrected it to MJ m-2 d-1 in the revised version, which is consistent with Figure 3 and commonly used in light use efficiency studies. Thank you for pointing it out! Fang, H., Wei, S., Jiang, C., & Scipal, K. (2012). Theoretical uncertainty analysis of global MODIS, CYCLOPES, and GLOBCARBON LAI products using a triple collocation method. Remote Sensing of Environment, 124, 610–621. https://doi.org/10.1016/j.rse.2012.06.013
3. Some other minor issues on the PAR estimation. Equation 9 uses the term "fPAR", which in this literature often means 'fraction of photosynthetically active radiation absorbed by plants.' (e.g, their equation 4). It's quite confusing to distinguish between fpar and FPAR. Finally, the authors might consider a supplemental figure showing patterns of PAR/SW ⅘AĚĞT this is a fairly well-studied, physically grounded ratio. It might also help to show that the approach works just fine in semi-cloudy (e.g, when there is more diffuse PAR) conditions.

We have followed all your suggestions in the revised manuscript. We have changed "fPAR" to "pPAR" (proportion of PAR in SW) to avoid misleading. We have added a PAR/SW ratio figure in the supplementary. We have also added a supplemental figure to show the error distribution of as a function of atmospheric transmittance (tSWR) to demonstrate that the approach works fine for any sky conditions.

4. For C3/C4, how are these uncertainties propagated into the final reported GPP estimates? From Figure 6b, it seems uncertainty is often quite high (e.g., > 40 percent). Accounting for this uncertainty seems important. For Figure 6 and 7, it could be helpful to change "Reference" to "CDL Reference" or something of that sort. At first, I was confused by the difference between "Reference" and "Ground Truth."

We agree that the uncertainties in C4 crop fraction is large (e.g., > 0.4) in some areas in Figure 6b. However, the metric RMSE is sensitive to extreme values, and it is different from misclassification rate (0.4 does not mean 40%). For a pure pixel of a corn/soybean rotation field, the RMSE = 0.39 if three out of 20 years is misclassified, i.e., misclassification rate = 0.15. For example, in Figure 7b, predicted time series match well with the CDL reference in most cases, but the RMSE is 0.40. We use RMSE instead of misclassification rate because C4 crop fraction is a numerical variable. To avoid misleading, we have added this clarification in the revised manuscript, and added RMSE values in Figure 7 to provide an intuitive sense of this uncertainty. We have also replaced "Reference" by "CDL reference" following your kind suggestion.

[Figure]

5. Perhaps most importantly, the authors mostly ignore the poor performance of their approach at evergreen needleleaf sites. Figure 8D indicates that SANIRv is not a good predictor of daily iPUE at ENF sites. The result is mentioned in L374, but relatively little discussion is offered for why this might be the case or what should be done about it. From Figure 11b, it seems that combining uncertain estimates of iPUE with PAR somewhat alleviates the poor performance within ENF, though the ENF site with an R2 of less than 0.2 stands out.

We totally agree with your point. We have added more interpretation on the poor performance in ENF: "This relatively weak iPUE $\sim$ SANIRv relationship is expected because evergreen needleleaf forest tends to allocate resources for leaf construction and maintenance at large time scales and does not have much flexibility to change canopy structure and leaf color as a response to varying environment at small time scales (Badgley et al., 2019; Chabot and Hicks, 1982). Previous studies found that changes in xanthophyll cycle instead of chlorophyll concentration or absorbed PAR explained the seasonal variation of photosynthetic capacity in evergreen needleleaf forest (Gamon et al., 2016; Magney et al., 2019). Therefore, SIF was suggested by some studies as a better proxy of photosynthetic capacity in this ecosystem (Smith et al., 2018; Turner et al., 2019), though satellite SIF has coarser spatial resolution, shorter temporal coverage, and larger temporal latency, and lower signal-to-noise ratio than SANIRv."

For the ENF site with the lowest R2 (US-Me1, Figure R1), we found it was likely a SANIRv data processing issue. First, this site has strong seasonal variation in NIRv and thus was not detected as ENF using our algorithm. Second, our soil correction uses 20 years data to derive soil background NIRv. At this site, NIRv data in 2004 – 2005 winter was much smaller than 20 years average, possibly due to poor growth conditions during that period as well as the contamination of snow or cloud. As a result, our algorithm over-corrected soil effect, leading to a lot of 0 SANIRv in that winter, which in the end caused low R2 in GPP. However, we considered this site-year as an extreme

case, which does not influence the overall quality of the product.

Figure R1 (see the attached figure). Comparison between AmeriFlux (black dots) and SLOPE (red curves) daily GPP at the US-Me1 site, which shows the lowest R2.

Badgley, G., Anderegg, L. D. ., Berry, J. A., & Field, C. B. (2019). Terrestrial Gross Primary Production: Using NIR V to Scale from Site to Globe. Global Change Biology, (April), 1–10. https://doi.org/10.1111/gcb.14729

Chabot, B. F., & Hicks, D. J. (1982). The ecology of leaf life spans. Annual Review of Ecology and Systematics. Volume 13, 13(1), 229–259. https://doi.org/10.1146/annurev.es.13.110182.001305

Magney, T. S., Bowling, D. R., Logan, B., Grossmann, K., Stutz, J., & Blanken, P. (2019). Mechanistic evidence for tracking the seasonality of photosynthesis with solar-induced fluorescence. Proceedings of the National Academy of Sciences, (27). https://doi.org/10.1073/pnas.1900278116

Gamon, J. A., Huemmrich, K. F., Wong, C. Y. S., Ensminger, I., Garrity, S., Hollinger, D. Y., . . . Peñuelas, J. (2016). A remotely sensed pigment index reveals photosynthetic phenology in evergreen conifers. Proceedings of the National Academy of Sciences, 201606162. https://doi.org/10.1073/pnas.1606162113

Smith, W. K., Biederman, J. A., Scott, R. L., Moore, D. J. P., He, M., Kimball, J. S., . . . Litvak, M. E. (2018). Chlorophyll Fluorescence Better Captures Seasonal and Interannual Gross Primary Productivity Dynamics Across Dryland Ecosystems of Southwestern North America. Geophysical Research Letters, 45(2), 748–757. https://doi.org/10.1002/2017GL075922

Turner, A. J., Köhler, P., Magney, T. S., Frankenberg, C., Fung, I., & Cohen, R. C. (2020). A double peak in the seasonality of California's photosynthesis as observed from space. Biogeosciences, 17(2), 405–422. https://doi.org/10.5194/bg-17-405-2020

6. A related, global scale effort to relate NIRv to GPP [Badgley, Anderegg, Berry,

Field, GCB2019] that the authors cite, identified 'deciduous' vs. 'evergreen' as being a critical parameter for model performance. Recognizing the difference in scales of the two analysis and the authors' stronger focus on C3/C4, it still feels necessary for a richer discussion of the performance of the model at ENF sites, especially given that the ultimate goal of the manuscript is to distribute a GPP dataset that researchers from across disciplines might find useful. From the analyses, it seems individuals working in agricultural contexts might find the data more reliable than those working in ENF systems. These caveats should be clearly flagged for the reader and the authors might benefit the research community by offering some discussion about what they think is going on and what future efforts might address such uncertainties.

(Badgley et al., 2019) separated 'evergreen' from 'deciduous' because they found "differing slopes for evergreen, deciduous, and crop ecosystem types". In our study, however, we found ENF has similar slope (c = 3.35) with all C3 PFTs (c = 3.46). Therefore, we decided not to separate them. We have made a richer discussion of the performance of the model at ENF sites (see response 5). In addition, we have further highlighted the limitation in the end of our Conclusion: "However, caution should be used in the interpretation of GPP seasonal trajectory in evergreen needleleaf forests because of relatively poor relationship between SANIRv and iPUE, and GPP magnitude in southwestern US grasslands because of the ignorance of fraction of C4 grasslands."

7. Finally, I find it perplexing that the manuscript lists five authors but the underlying data product only lists two authors. The author contributions indicate that author G.W. helped develop the SLOPE model and that authors B.P. and S.W. were involved in refining/interpreting how the model works. Doesn't the resulting dataset and the citations it might one day receive rely on the contributions of these three authors as well?

We considered model development, paper writing, and product generation as different works. Therefore, paper and product have different author lists.

[Figure]

2020.

[Figure]

[Figure]

**Fig. 1.** Figure R1. Comparison between AmeriFlux (black dots) and SLOPE (red curves) daily GPP at the US-Me1 site, which shows the lowest R2.

---

## Author Comment (AC2) · 1 Oct 2020

Thank you for reviewing our manuscript "A daily, 250 Ĺm, and real-time gross primary productivity product (2000–present) covering the Contiguous United States". We have tried our best to address your comments and to improve our manuscript.

1. This manuscript details a new SateLite Only Photosynthesis Estimation (SLOPE) gross primary productivity (GPP) product based on: 1) near-infrared reflectance of vegetation (NIRv); 2) photosynthetically active radiation (PAR); and 3) C3/C4 fractional cover. The new product explains 84% of the spatial and temporal variance in GPP obtained from 50 Ameriflux eddy covariance flux tower (EC) sites. Critically, the prod-

uct includes uncertainty estimates at the pixel-level, an important advance over most existing products.

We greatly appreciate your positive summary on our manuscript.

2. Overall, I feel the paper is well written and the data product is of significant value, but perhaps mostly as an improved proxy for cropland productivity. The authors make importance advances over previous efforts by incorporating satellite-based NIRv and PAR and removing dependence on reanalysis-based weather data. The use of C3 and C4 fractional cover is appropriately done for croplands, but importantly it does not appear appropriately handled for natural ecosystems. The work adapts the commonly used light use efficiency framework, but removes all biophysical constraint logic (e.g., response to temperature, water, nutrient limitation), and instead makes the assumption that NIRv adequately captures these constraints. While this might be a fair assumption for herbaceous and deciduous dominated ecosystems, it is likely problematic for natural evergreen dominated ecosystems. Additionally, independent of vegetation type, it is unclear if NIRv is capable of capturing changes in LUE such that CO2 fertilization effects are accurately represented in this product. Further, the authors utilize a classification that separates C3 and C4 vegetation functional types, however, their input data does not separate natural C3 and C4 grasslands, which is likely problematic for western US ecosystems (which are under-represented by eddy covariance flux towers, and thus the product is not well evaluated across these regions). In my view, these critical issues need to be fully addressed before this manuscript can be considered for publication.

Thank you for providing these deep insights. We totally agree with all of your points. First, SLOPE is developed to generate a high-spatiotemporal-resolution real-time GPP dataset with a reasonable overall accuracy. This objective fundamentally differentiates it from existing GPP products. In fact, we believe process-based model is the best way to quantify GPP, because it takes all influential factors (e.g., temperature, water supply and demand, radiation quantity and quality, CO2 fertilization, nutrient limitation, leaf

physiology, and canopy structure) into consideration. The authors have been persistently devoting to this direction (Jiang and Ryu, 2016; Jiang et al., 2020). However, we are aware that large uncertainties accumulate from both parameterizations and data, and therefore not suitable to generate a high-spatiotemporal-resolution real-time GPP dataset with a reasonable overall accuracy. Light use efficiency model removes heavy burdens from process-based model and is therefore more practical with a reasonable overall accuracy, but it also removes many useful features. Most LUE models only use information content in PAR, FPAR, temperature and humidity, without considerations of $CO_2$ fertilization and nutrient limitation. The parameterizations of temperature and humidity effects are usually empirical, and brings in low resolution and large latency input data. SLOPE removes these uncertainties in both parameterizations and data through removing explicit biophysical constraints. Because plants adapt to environment and optimize their canopy structure and leaf physiology, NIRV,Ref is likely to capture a majority part of biophysical constraints. However, we acknowledge that where and when this strategy leads to more benefits requires much more investigations in the future.

Second, we totally agree with your point on ENF. We have added more interpretation on the poor performance in ENF: "This relatively weak iPUE $\sim$ SANIRv relationship is expected because evergreen needleleaf forest tends to allocate resources for leaf construction and maintenance at large time scales and does not have much flexibility to change canopy structure and leaf color as a response to varying environment at small time scales (Badgley et al., 2019; Chabot and Hicks, 1982). Previous studies found that changes in xanthophyll cycle instead of chlorophyll concentration or absorbed PAR explained the seasonal variation of photosynthetic capacity in evergreen needleleaf forest (Gamon et al., 2016; Magney et al., 2019). Therefore, SIF was suggested by some studies as a better proxy of photosynthetic capacity in this ecosystem (Smith et al., 2018; Turner et al., 2019), though satellite SIF has coarser spatial resolution, shorter temporal coverage, and larger temporal latency, and lower signal-to-noise ratio than SANIRv."

Third, it is true that the missing of C4 grasses is a limitation. We had a statement in the manuscript that "It is worth mentioning that C4 grassland and shrubland are not considered in this study as no nationwide high-resolution distribution data is available". We have further highlighted the limitation in the end of our Conclusion: "However, caution should be used in the interpretation of GPP seasonal trajectory in evergreen needle-leaf forests because of relatively poor relationship between SANIRv and iPUE, and GPP magnitude in southwestern US grasslands because of the ignorance of fraction of C4 grasslands."

Jiang, C., & Ryu, Y. (2016). Multi-scale evaluation of global gross primary productivity and evapotranspiration products derived from Breathing Earth System Simulator (BESS). Remote Sensing of Environment, 186, 528–547. https://doi.org/10.1016/j.rse.2016.08.030

Jiang, C., Ryu, Y., Wang, H., & Keenan, T. F. (2020). An optimality-based model explains seasonal variation in C3 plant photosynthetic capacity. Global Change Biology, gcb.15276. https://doi.org/10.1111/gcb.15276

Badgley, G., Anderegg, L. D. ., Berry, J. A., & Field, C. B. (2019). Terrestrial Gross Primary Production: Using NIR V to Scale from Site to Globe. Global Change Biology, (April), 1–10. https://doi.org/10.1111/gcb.14729

Chabot, B. F., & Hicks, D. J. (1982). The ecology of leaf life spans. Annual Review of Ecology and Systematics. Volume 13, 13(1), 229–259. https://doi.org/10.1146/annurev.es.13.110182.001305

Magney, T. S., Bowling, D. R., Logan, B., Grossmann, K., Stutz, J., & Blanken, P. (2019). Mechanistic evidence for tracking the seasonality of photosynthesis with solar-induced fluorescence. Proceedings of the National Academy of Sciences, (27). https://doi.org/10.1073/pnas.1900278116

Gamon, J. A., Huemmrich, K. F., Wong, C. Y. S., Ensminger, I., Garrity, S., Hollinger,

D. Y., ... Peñuelas, J. (2016). A remotely sensed pigment index reveals photosynthetic phenology in evergreen conifers. Proceedings of the National Academy of Sciences, 201606162. https://doi.org/10.1073/pnas.1606162113

Smith, W. K., Biederman, J. A., Scott, R. L., Moore, D. J. P., He, M., Kimball, J. S., ... Litvak, M. E. (2018). Chlorophyll Fluorescence Better Captures Seasonal and Interannual Gross Primary Productivity Dynamics Across Dryland Ecosystems of Southwestern North America. Geophysical Research Letters, 45(2), 748–757. https://doi.org/10.1002/2017GL075922

Turner, A. J., Köhler, P., Magney, T. S., Frankenberg, C., Fung, I., & Cohen, R. C. (2020). A double peak in the seasonality of California's photosynthesis as observed from space. Biogeosciences, 17(2), 405–422. https://doi.org/10.5194/bg-17-405-2020

3. Line 50-60: Some valid points are made here. However, for the CONUS region in particular, there have been previous advances that already address many of these limitations. In particular, Robinson et al. (2018) utilized high quality weather data interpolated from dense weather station networks across the region and improved landcover data from the National Landcover Data Layer (NLCD). This is a much more appropriate data product to compare the new product against and, if it's available, I recommend this comparison.

Thank you for your kind suggestion. First, we have changed "50 km" to "> 10 km" in the revised manuscript. Second, NLCD still suffers from large time lag issue. Third, we cannot agree more on the importance to compare different products. We suppose comparison should be made in a comprehensive manner, as the authors did in the past (Jiang et al., 2016). However, this is too much for this manuscript and also is not encouraged by this journal. We plan to do that in a separate study.

4. Sections 3.1 and 3.2: I commend the authors for their work to provide robust uncertainty estimates for PAR and SANIRv, and subsequently SLOPE GPP estimates. This is an important advance over most previous efforts. Is the uncertainty due to PAR,

SANIRv, functional classification, and statistical fit included as separate data layers with the final product? This would be a useful addition for understanding pixel-level uncertainty. Also it would be useful to see a map that shows the dominant source of uncertainty at the pixel level, which would highlight potentially how uncertainty varies by region.

We really appreciate this highlight on our novelty. However, we are aware that the uncertainty definitions of different components are quite different: • PAR: "Four different PAR estimations are then obtained by Eq. (9), and their ensemble mean and standard deviation are considered as the final estimation and uncertainty, respectively". • SANIRv: "SANIRV is supposed to be smooth within a short time period, therefore, the standard deviation within the ±3-day temporal window is calculated as uncertainty". • C4 crop fraction: "The RMSE between predicted and reference CDL C4 fraction is calculated as uncertainty". • Slope coefficient: "The RMSE between SANIRV-derived and AmeriFlux iPUE for C3 and C4 are calculated as uncertainties of cC3 and cC4, respectively". Although we integrated them to provide a quantitative GPP uncertainty, the attribution to different components and the analysis of their relative importance may not make enough sense. Rather, we suppose investigating the spatiotemporal patterns of GPP product uncertainty makes a lot of sense. However, this requires a comprehensive analysis, as the authors did in the past (Fang et al., 2013), which is too much for this manuscript.

Fang, H., Jiang, C., Li, W., Wei, S., Baret, F., Chen, J. M., . . . Zhu, Z. (2013). Characterization and intercomparison of global moderate resolution leaf area index (LAI) products: Analysis of climatologies and theoretical uncertainties. Journal of Geophysical Research: Biogeosciences, 118(2), 529–548. https://doi.org/10.1002/jgrg.20051

5. Section 3.2: There is very low variance in SANIRv for evergreen vegetation (Figure 5), which results in the lowest correlation with EC GPP data. There are also a few ENF sites with very low correlation (Figure 11). This has been a well-known issue of vegetation reflectance-based indices, such as NIRv. Below are a number of papers that

discuss this issue and all indicate SIF could be a major improvement. I recommend at minimum that this issue and ways forward, such as downscaling TROPOMI SIF (Turner et al., 2019), be included prominently in the discussion of this paper.

Thank you for your suggestion. Please refer to our response 2.

6. Section 3.3: There are dynamic mixtures of C3 and C4 species throughout the natural ecosystems of the Western US. As far as I can tell this analysis and it's reliance on NLCD data is unable to account for these important ecosystems since NLCD does not represent C3 and C4 grasslands. These are also regions where EC sites are not well represented and thus the product 1) does not accurately capture; and 2) is not well constrained or evaluated across these ecosystems. At minimum, this needs to be pointed out very clearly throughout the methods and discussion of this paper. Alternatively, and maybe more appropriately, it seems the authors present an advanced cropland productivity product and perhaps natural regions including C3/C4 grasslands and evergreen forests should be masked out. It is my view that the current work has major limitations for accurately representing natural ecosystems.

Thanks for pointing out this issue. We were aware of the paper you suggested. However, the dataset and model generated in that paper was limited in a small region and we were unable to apply it to the whole CONUS. We had a statement in the manuscript that "It is worth mentioning that C4 grassland and shrubland are not considered in this study as no nationwide high-resolution distribution data is available". We have further highlighted the limitation in the end of our Conclusion: "However, caution should be used in the interpretation of GPP seasonal trajectory in evergreen needleleaf forests because of relatively poor relationship between SANIRv and iPUE, and GPP magnitude in southwestern US grasslands because of the ignorance of fraction of C4 grasslands."

7. The authors cite Badgley et al. (2019) as justification for separating the model based on only C3 and C4 functional types. Yet, that paper also indicated that the

best model fit between NIRv and GPP included separation between deciduous and evergreen ecosystem types as well. Based on this, the authors should include further justification for their model framework.

(Badgley et al., 2019) separated 'evergreen' from 'deciduous' because they found "differing slopes for evergreen, deciduous, and crop ecosystem types". In our study, however, we found ENF has similar slope ($c = 3.35$) with all C3 PFTs ($c = 3.46$). Therefore, we decided not to separate them.

8. I recommend a map of the flux sites utilized overlaying the NLCD / CDL data utilized. This would highlight that the product has not been evaluated across important western US ecosystem types including dry herbaceous, shrub, and evergreen forests. Also for Table S1, one of the only dryland evergreen forest sites utilize (NR1) is reproduced twice and it's unclear if years 2000-2007 or 2000-2014 are utilized in the analysis.

Thank you so much for point out this fault. Yes, we double-counted this site (US-NR1). We used two different GPP datasets, AmeriFlux L4 and FLUXNET2015. We gave priority to FLUXNET2015 and removed the overlapped sites from AmeriFlux L4. We have remade this table as well as all results. We have highlighted the limitation in the end of our Conclusion: "However, caution should be used in the interpretation of GPP seasonal trajectory in evergreen needleleaf forests because of relatively poor relationship between SANIRv and iPUE, and GPP magnitude in southwestern US grasslands because of the ignorance of fraction of C4 grasslands."

9. Consider dropping the NASA Blue Marble background from all maps. This is unnecessary and potentially distracting.

We have followed your suggestion and remade all maps.

10. Why use a soil adjusted NIRv? One of the advantages of NIRv is that is naturally isolates a pure vegetation signal (Bagdley et al., 2019).

Because NIRv is still influenced by soil signal. We had a statement on this concern:
"When vegetation is absent, iPUE is zero and NIRV,Ref should be zero too. However, this is not true in reality as >99.9% soils have positive NIRV,Ref values according to a global soil spectral library (Jiang and Fang, 2019), and the correction of NIRV,Ref for soil is needed for better performance at low vegetation cover (Zeng et al., 2019)." By correcting the soil effect, the intercept term can be removed from the iPUE $\sim$ NIRV,Ref relationship.

11. I recommend including in the discussion whether this model is capable of capturing $CO_2$ fertilization effects on vegetation productivity. Previous work has suggested that LUE models may not have the capacity to fully capture this important and rapidly changing driver of GPP. Pointing out this potential limitation is important to ensure appropriate data usage by the community.

This is a great point. We have added one sentence in Conclusion: "Although the SLOPE product has been generated from 2000 to present, caution should be used in the interpretation of long-term trend because the SLOPE model, as many other LUE models, does not explicitly consider the $CO_2$ fertilization effects on vegetation productivity."

---

## Author Comment (AC3) · 1 Oct 2020

Thank you for reviewing our manuscript "A daily, 250 m, and real-time gross primary productivity product (2000–present) covering the Contiguous United States". We have tried our best to address your comments and to improve our manuscript.

1. This study proposes a new diagnostic estimation of gross primary productivity at a continental scale using remote sensing data. The proposed method is good for many researchers, etc., since it provides high spatial (250m) and temporal (daily) resolutions. Another advantage of this approach is its simplicity (a very small number of parameters). Also, model description, evaluation, and data availability were well-written, I

believe. I have some minor comments (and questions).

We greatly appreciate your positive comments.

2. Uncertainties estimation. Adding uncertainty information is very nice, and one of the most important contributions of this study. I found clear description on GPP uncertainties, however, it is not easy to find information on each input parameters. Please add information how to define uncertainties in each input data.

Uncertainty of PAR estimation is described at L159–160 in the original manuscript: "Four different PAR estimations are then obtained by Eq. (9), and their ensemble mean and standard deviation are considered as the final estimation and uncertainty, respectively". Uncertainty of SANIRv estimation is described at L212–213 in the original manuscript: "SANIRV is supposed to be smooth within a short time period, therefore, the standard deviation within the $\pm$3-day temporal window is calculated as uncertainty". Uncertainty of C4 fraction estimation is described at L244–245 in the original manuscript: "The RMSE between predicted and reference CDL C4 fraction is calculated as uncertainty". Uncertainty of slope coefficient is described at L271–272 in the original manuscript: "The RMSE between SANIRV-derived and AmeriFlux iPUE for C3 and C4 are calculated as uncertainties of cC3 and cC4, respectively".

L52-53. As far as I heard, there are several new reanalysis datasets for meteorological datasets with higher spatial resolutions. Please update the description here. I think this approach is not based on GCM, but reanalysis (of course, reanalysis is based on climate model, but reanalysis is more appropriate word. Please modify.

Thanks for the suggestions. We have changed "50 km" to "> 10 km". We have replaced "GCMs" by "reanalysis approaches".

L77. Maybe no explanation of a symbol, REDref. Please check the manuscript.

Thank you for the remaindering. We have added it to the sentence "MODIS provides long-term and real-time (2000 – present) observations of red (RedRef) and NIR (NIR-

Ref) reflectance".

L97. 'When vegetation in absent, iPUE is zero, and NIRvref should be zero." I don't think NIRref should be zero, since soil shows different NIRvref value. Suggest rewording.

NIRv,ref is near-infrared reflectance of vegetation. We have revised it to "When vegetation is absent, iPUE is zero and NIRV,Ref is expected to be zero".

L166-169. For quality control of surface reflectance data, did you use information on MOD09(MYD09) quality flag information? I could not find the description. If no, why?

Thanks for this comment. We did apply QC. We have added one sentence: "Only pixels with QC information of 'corrected product produced at ideal quality all bands' were used."

L176. Sensor view angle is a cause of Terra, Aqua reflectance differences. Solar zenith and azimuth angles are also important as well as sensor view angle?

Thanks for the suggestion. Terra's average overpass is around 11am, whereas Aqua's is around 1pm. They have similar solar zenith angles, but could differ in solar azimuth angle. We have revised this sentence: "the remaining cloud contamination and sun-target-sensor geometry could cause differences between morning and afternoon observations".

L196-207. Not clear, please improve it to understand it more easily.

We have revised it as follows: "For a specific pixel, soil background NIRV (NIRV,Soil) is supposed to 1) smaller than seasonal mean NIRV,Ref which includes vegetated period, and 2) smaller than 0.2 indicated by a global soil spectral library (Jiang and Fang, 2019). Therefore, NIRV,Soil is supposed to within a range of [0, min(mean(NIRV,Ref), 0.2)]. The mode of daily NIRV,Ref averaged over 2000 – 2019 within this value range is considered as NIRV,Soil. An example is given in Figure S5. Theoretically, NIRV,Soil for evergreen species cannot be obtained from time series NIRV,Ref because of the

persistent vegetation cover. Pixels with NIRV,Soil value larger than 0.1 and seasonal coefficient of variation (CV) of NIRV,Ref smaller than 33% are empirically considered as evergreen species, and their NIRV,Soil values are set to 0."

---

## Author Response (AR1)

Thank you for reviewing our manuscript "A daily, 250 m, and real-time gross primary productivity product (2000–present) covering the Contiguous United States". We have tried our best to address your comments and to improve our manuscript.

Reviewer #1

1. The authors have wrapped three separate innovations into a single paper. First, a new approach to estimating PAR from MODIS. Second, a new approach for estimating C3/C4 fraction. And third, combining those two new products with estimates of NIRv to derive GPP.

We greatly appreciate your positive summary on our innovations.

2. For the PAR modeling, either more information is needed about the "four machine learning approaches" or the authors need to provide the code used to fit these models. Each algorithm has a multitude of adjustable parameters. How were these parameters determined, what are their values, and how might future researchers modify and/or improve upon the approach? It is also unclear to me how "uncertainty" is calculated. As written (L160), it appears uncertainty is represented in terms of model-model disagreement, as opposed to model-data disagreement. This seems inappropriate, though I know that the approach has been used in other parts of this literature. Trying to wrap my head around the uncertainty terms was complicated by the fact that Figure 2 reports PAR in W m2, while Figure 3 uses MJ m2 d1. Using common units throughout would be helpful.

We totally agreed with your points.

For the modeling, we have added the following text to the manuscript (L157):

"We used Scikit-learn, a free software machine learning library for the Python programming language, to build the models. All the four algorithms were automatically optimized by tuning their hyperparameters using five-fold-cross-validation on their training dataset."

For the uncertainty, we suppose product uncertainty information can be categorized into two types: theoretical and physical (Fang et al., 2012). While physical uncertainties indicate the departure of product values from hypothetical true values and are obtained through independent validation studies, i.e., model-data disagreement, theoretical uncertainties are caused by uncertainties in the input data and model imperfections, e.g., model-model disagreement, and are usually estimated by individual product science teams. The unit in Figure 2 is a typo, and we have corrected it to MJ m-2 d-1 in the revised version (L275), which is consistent with Figure 3 and commonly used in light use efficiency studies. Thank you for pointing it out!

Fang, H., Wei, S., Jiang, C., & Scipal, K. (2012). Theoretical uncertainty analysis of global MODIS, CYCLOPES, and GLOBCARBON LAI products using a triple collocation method. *Remote Sensing* of Environment, 124, 610–621. https://doi.org/10.1016/j.rse.2012.06.013 3. Some other minor issues on the PAR estimation. Equation 9 uses the term "fPAR", which in this literature often means 'fraction of photosynthetically active radiation absorbed by plants.' (e.g, their equation 4). It's quite confusing to distinguish between fpar and FPAR. Finally, the authors might consider a supplemental figure showing patterns of PAR/SW  $\hat{a}$ 'A'T this is a fairly well-studied, physically grounded ratio. It might

**also help to show that the approach works just fine in semi-cloudy (e.g, when there is more diffuse PAR) conditions.**

We have followed all your suggestions in the revised manuscript. We have changed " $f_{PAR}$ " to " $p_{PAR}$ " (proportion of PAR in SW) to avoid misleading (e.g., L146). We have added a PAR/SW ratio figure in the supplementary (Figure S5). We have also added a supplemental figure to show the error distribution of as a function of atmospheric transmittance (SWR/SWRTOA) to demonstrate that the approach works fine for any sky conditions (Figure S6).

**L282:**

"In addition to the total amount of PAR, SLOPE PAR also reveals variations in the ratio of PAR to SWR (Fig. S5). Despite of a relatively small range (0.40–0.46), it is negatively correlates with cloud optical thickness and total ozone burden and positively correlates with total water vapor."

Figure S5. Spatial distribution of PAR/SWR on Jul 10, 2020, and its spatial relationship with COT, AOD, TWV and TO3.

**L295:**

In addition, the performance is reasonably stable under different sky conditions, indicated by similar R2 and RMSE values from low to high atmospheric transmittance (Figure S6).

Figure S6.  $R^2$  and RMSE of SLOPE PAR as a function of atmospheric transmittance (SWR/SWRTOA) across seven SURFRAD sites in 2019.

4. For C3/C4, how are these uncertainties propagated into the final reported GPP estimates? From Figure 6b, it seems uncertainty is often quite high (e.g., > 40 percent). Accounting for this uncertainty seems important. For Figure 6 and 7, it could be helpful to change "Reference" to "CDL Reference" or something of that sort. At first, I was confused by the difference between "Reference" and "Ground Truth."

We agree that the uncertainties in C4 crop fraction is large (e.g., > 0.4) in some areas in Figure 6b. However, the metric RMSE is sensitive to extreme values, and it is different from misclassification rate (0.4 does not mean 40%). For a pure pixel of a corn/soybean rotation field, the RMSE = 0.39 if three out of 20 years is misclassified, i.e., misclassification rate = 0.15. For example, in Figure 7b, predicted time series match well with the CDL reference in most cases, but the RMSE is 0.40. We use RMSE instead of misclassification rate because C4 crop fraction is a numerical variable. To avoid misleading, we have added this clarification in the revised manuscript, and added RMSE values in Figure 7 to provide an intuitive sense of this uncertainty. We have also replaced "Reference" by "CDL reference" following your kind suggestion.

**L341:**

"It is worth mentioning that the uncertainty metric RMSE is sensitive to extreme values, and it is different from misclassification rate (0.4 does not mean 40%). For a pure pixel of a corn/soybean rotation field, the RMSE = 0.39 if three out of 20 years is misclassified, i.e., misclassification rate = 0.15."

5. Perhaps most importantly, the authors mostly ignore the poor performance of their approach at evergreen needleleaf sites. Figure 8D indicates that SANIRv is not a good predictor of daily iPUE at ENF sites. The result is mentioned in L374, but relatively little discussion is offered for why this might be the case or what should be done about it. From Figure 11b, it seems that combining uncertain estimates of iPUE with PAR somewhat alleviates the poor performance within ENF, though the ENF site with an R2 of less than 0.2 stands out.

We totally agree with your point. We have added more interpretation on the poor performance in ENF (L388):

"This relatively weak iPUE ~ SANIRv relationship is expected because evergreen needleleaf forest tends to allocate resources for leaf construction and maintenance at large time scales and does not have much flexibility to change canopy structure and leaf color as a response to varying environment at small time scales (Badgley et al., 2019; Chabot and Hicks, 1982). Previous studies found that changes in xanthophyll cycle instead of chlorophyll concentration or absorbed PAR explained the seasonal variation of photosynthetic capacity in evergreen needleleaf forest (Gamon et al., 2016; Magney et al., 2019). Therefore, SIF was suggested by some studies as a better proxy of photosynthetic capacity in this ecosystem (Smith et al., 2018; Turner et al., 2019), though satellite SIF has coarser spatial resolution, shorter temporal coverage, and larger temporal latency, and lower signal-to-noise ratio than SANIRv."

For the ENF site with the lowest  $R^2$  (US-Me1, Figure R1), we found it was likely a SANIRv data processing issue. First, this site has strong seasonal variation in NIRv and thus was not detected as ENF using our algorithm. Second, our soil correction uses 20 years data to derive soil background NIRv. At this site, NIRv data in 2004 – 2005 winter was much smaller than 20 years average, possibly due to poor growth conditions during that period as well as the contamination of snow or cloud. As a result, our algorithm over-corrected soil effect, leading to a lot of 0 SANIRv in that winter, which in the end caused low  $R^2$  in GPP. However, we considered this site-year as an extreme case, which does not influence the overall quality of the product.